# Association of Caspase 3 Activation and H2AX γ Phosphorylation in the Aging Brain: Studies on Untreated and Irradiated Mice

**DOI:** 10.3390/biomedicines9091166

**Published:** 2021-09-06

**Authors:** Nadia Gionchiglia, Alberto Granato, Adalberto Merighi, Laura Lossi

**Affiliations:** Department of Veterinary Sciences, University of Turin, 10095 Grugliasco (TO), Italy; nadia.gionchiglia@unito.it (N.G.); alberto.granato@unito.it (A.G.); adalberto.merighi@unito.it (A.M.)

**Keywords:** aging, forebrain, neurons, H2AX, ionizing radiations, DNA, DNA damage, cell proliferation, apoptosis, caspase 3

## Abstract

Phosphorylation of H2AX is a response to DNA damage, but γH2AX also associates with mitosis and/or apoptosis. We examined the effects of X-rays on DNA integrity to shed more light on the significance of H2AX phosphorylation and its relationship with activation of caspase 3 (CASP3), the main apoptotic effector. After administration of the S phase marker BrdU, brains were collected from untreated and irradiated (10 Gray) 24-month-old mice surviving 15 or 30 min after irradiation. After paraffin embedding, brain sections were single- or double-stained with antibodies against γH2AX, p53-binding protein 1 (53BP1) (which is recruited during the DNA damage response (DDR)), active CASP3 (cCASP3), 5-Bromo-2-deoxyuridine (BrdU), and phosphorylated histone H3 (pHH3) (which labels proliferating cells). After statistical analysis, we demonstrated that irradiation not only induced a robust DDR with the appearance of γH2AX and upregulation of 53BP1 but also that cells with damaged DNA attempted to synthesize new genetic material from the rise in BrdU immunostaining, with increased expression of cCASP3. Association of γH2AX, 53BP1, and cCASP3 was also evident in normal nonirradiated mice, where DNA synthesis appeared to be linked to disturbances in DNA repair mechanisms rather than true mitotic activity.

## 1. Introduction

The participation of DNA damage and repair in aging processes is well recognized. Aging is an undisputable risk factor for neurodegenerative diseases, emphasizing the importance of research into the role that DNA modifications play in the genesis of these disorders.

The preservation of genomic integrity is essential for physiological cellular functions. As mature neurons are postmitotic cells, where DNA damage rises with age and may worsen in the presence of neurological disorders, it is difficult to maintain such integrity over a lifetime. Therefore, the border between physiological and pathological aging is quite elusive.

Several types of damage can affect DNA, among which single- (SSBs) and double-stranded breaks (DSBs) are the most frequently observed. SSBs chiefly derive from DNA topoisomerase abortive activity, reactive oxygen species (ROS) attacks, and several other DNA damaging substances. Additionally, they arise when base excision repair (BER) is active. If SSBs are not repaired, potential consequences include replication fork collapse or conversion to DSBs [1]. DSBs are the main cytotoxic lesion for ionizing radiations and radio-mimetic chemicals. They can also be caused by mechanical stress on chromosomes or when a replicative DNA polymerase encounters an SSB or other types of DNA lesions. DSBs can also derive from apoptosis-associated DNA fragmentation [2].

SSBs and DSBs have different degrees of severity. DSBs are relatively rare but have a strong impact on neurodevelopment as they undermine the integrity of proliferating and differentiating cells, leading to an array of disorders from embryonic lethality to several forms of ataxia [3]. In contrast, SSBs, which are three orders of magnitude more frequent, are less severe as they undergo repair very quickly and are unlikely to cause developmental defects or microcephaly. Nevertheless, in postmitotic neurons, SSBs can be responsible for several forms of progressive neurodegeneration and cerebellar ataxia [3].

Another difference between SSBs and DSBs is their propensity to provoke apoptosis. SSBs are more likely to cause apoptosis, because the accumulation of p53 (one of the several molecules involved in repairing damaged DNA) is faster than in DSBs and, consequently, cells with SSBs undergo apoptosis more easily and have a lower apoptotic threshold than cells with DSBs [4].

Phosphorylation of the histone H2AX at the serine-139 position (γH2AX form) is an early response to a broad range of DNA damages. As a result, the generation of γH2AX is one of the earliest events in the DNA damage response (DDR) and is pivotal in sensing and repairing the injured DNA [5].

Previous in vitro studies in cells other than neurons—e.g., HL60, FL, CHL, BEAS-2B, Jurkat, and HCT116 cell lines—have employed γH2AX immunofluorescence (IMF) in combination with analysis of the cellular DNA content, chromatin condensation, and caspase 3 (CASP 3) activation to distinguish between the γH2AX fluorescence derived from different types of DNA damage or apoptosis-associated DNA fragmentation [2,6]. These studies demonstrated that when apoptotic cells were immunostained with an anti-γH2AX antibody, pan-nuclear diffuse staining occurred, instead of the γH2AX foci that are typically associated with the DDR.

As regards the connections between γH2AX and CASP3, the main effector protease in apoptosis [7], observations have been very rare in neurons and remain controversial in other cell types. An initial study in non-neural HL60 cells showed that induction of γH2AX in response to DNA lesions preceded apoptosis [8]. However, another survey reported that H2AX was phosphorylated after activation of the apoptotic machinery in several human cell lines of different origins [9]. Studies in neurons, on the other hand, have not primarily focused on the links between H2AX and apoptosis. They aimed, instead, to assess the response of isolated cells or the intact brain to several DNA-damaging agents of heterogeneous nature (e.g., hydrogen peroxide [10] and mifepristone [11]) that have been reported to induce γH2AX with [11] or without [10] the concomitant activation of CASP3.

Analysis of the intact brain is more complicated than studying homogenous populations of isolated cells. Differing from neoplastic cells, many mature neurons enter a quiescent state of the cell cycle that is commonly referred to as the G_0_ phase [12]. In mice of different ages (from embryonic day 14.5 to 24 months) we previously showed that, after experimental labeling of DNA-synthesizing cells with 5-Bromo-2-deoxyuridine (BrdU), focal and/or diffuse γH2AX immunostaining appeared in interkinetic nuclei, mitotic chromosomes, and apoptotic nuclei [13]. The subventricular zone (SVZ) of the telencephalon and, to a much lesser extent, the subgranular zone of the hippocampal dentate gyrus were highly immunoreactive up to adulthood (about two months). Thereafter, up to 24 months, γH2AX remained highly expressed only in the cerebral cortex and, to a lesser extent, the SVZ/rostral migratory stream/olfactory bulb (SVZ/RMS/OB) system. Double-labeling experiments demonstrated that γH2AX expression in embryos, postnatal, adult, and old neurogenic brain areas correlated temporally and functionally to both proliferation and apoptosis of neuronal precursors. Conversely, in old mice, γH2AX-immunoreactive cortical neurons that incorporated BrdU did not express the proliferation marker phosphorylated histone H3 (pHH3), indicating that these postmitotic cells endured a substantial DDR in the absence of any known genotoxic event.

Because γH2AX induction not only precedes [8] but also follows apoptosis [9], naturally occurring neuronal death or pathological events leading to cell death could be a confounding factor in correctly interpreting the expression of γH2AX in the brain. Moreover, a very recent study demonstrated that uniform widespread nuclear phosphorylation of histone H2AX was an indicator of lethal DNA replication stress, rather than DDR [14].

Cell proliferation and apoptosis may occur in spatial and temporal overlapping windows during prenatal and postnatal neurogenesis, and the relationship of the two events remains an issue of debate regarding neurogenesis in adult and old mammals, including humans [15]. One of several markers of apoptosis is cleaved CASP3 (cCASP3), the most important effector caspase. The fully functional protease derives from an inactive precursor that becomes active by cleavage as the endpoint of a cascade of events ultimately leading to apoptotic death [16].

In our previous study, we identified apoptotic cells simply on their appearance at the light and electron microscopic levels [13]. To the best of our knowledge, there are no data in the literature correlating the expression of γH2AX and cCASP3 in the aging mouse brain. To better understand this correlation and its functional significance, we studied the expression of the cleaved protease in the brain of 24-month-old mice under normal conditions and after X-ray irradiation, which has major DNA damaging effects.

We first wanted to establish whether γH2AX was indeed a useful marker of DNA damage in the untreated mouse brain. To do so—as a positive control, taking advantage of the well-characterized effects of ionizing radiations onto the DNA—we experimentally injured the DNA with X-rays (Figure 1) and followed the onset and course of expression of γH2AX together with that of the p53-binding protein 1 (53BP1), which is involved in the early DNA damage-signaling pathway [17].

We also wanted to determine whether H2AX phosphorylation was associated with the activation of CASP3 at the single-cell level. Additionally, we intended to shed light on the functional relationship between phosphorylation of H2AX, activation of CASP3, and DNA synthesis, as detected after BrdU incorporation. Lastly, we investigated—again, at the single-cell level—the connection between BrdU incorporation and mitosis.

We first demonstrated that irradiation caused a robust DDR with intense phosphorylation of H2AX and upregulation of 53BP1. Then, we observed that cells with damaged DNA attempted to synthesize new genetic material, which was evident from the rise in BrdU immunostaining, with an increase in the expression of cCASP3. Notably, the association of γH2AX, 53BP1, and cCASP3 was also evident in normal nonirradiated mice, where DNA synthesis appeared to be linked to disturbances in DNA repair mechanisms rather than to true mitotic activity.

## 2. Materials and Methods

### 2.1. Animals and X-ray Irradiation

We performed our studies using ten 24-month-old CD1 mice and six 24-month-old B6/129 mice. All experiments were approved by the Italian Ministry of Health (authorizations n. 65/2016 PR of March 2016 to the University of Turin and n. 130/2012-B of January 2012 to Enea, Rome) and the Bioethics Committee of the University of Turin. X-ray irradiation of B6/129 mice was carried out at Enea in the framework of an authorized project aiming to study the genetic risk consequent to cancer radiotherapy. Mice brains that were unnecessary to that project were kindly donated for our purposes. All experiments were carried out according to the guidelines and recommendations of the European Union (Directive 2010/63/UE) as implemented by current Italian regulations on animal welfare (DL n. 26-04/03/2014). In keeping with the 3Rs principle, we kept the number of mice to the minimum necessary for statistical significance and made all efforts to minimize animal suffering during the sacrifice.

To study the link between DNA synthesis and γH2AX expression in normal and X-ray irradiated mice we used the following experimental protocol (Figure 1). We divided mice into three groups of two animals each: control, short survival, and long survival. At the beginning of the experiments, all animals received an intraperitoneal injection of 0.1 mg/g body weight BrdU (Sigma Chemicals, St. Louis, MO, USA) dissolved in sterile water. Mice were euthanized after two hours with an intraperitoneal injection of sodium pentobarbital (>30 mg/kg) without any further treatment (control) or after having been irradiated with a 10 Gray dose starting 15 (short survival) or 30 min before suppression (long survival). X-ray irradiation was performed with a Gilardoni CHF 320G X-ray generator (Gilardoni S.p.A., Mandello del Lario, Italia) on free-moving animals that were put under the radiation beam in a small plastic box. All animals belonging to the same experimental group were irradiated together. According to the features of the X-ray machine, administration of a 10 Gray dose of irradiation lasted 10 min.

### 2.2. Brain Sampling and Tissue Processing

Mice were perfused with 4% paraformaldehyde in 0.1 M phosphate buffer (PB). After dissection, brains were left in fixative for two additional hours. Paraffin embedding was carried out according to standard procedures. The brains were then serially sectioned (10 µm) along the parasagittal axis and all sections were collected on slides. Equally spaced sections were assigned to different series for specific single and double immunolabeling procedures (see Section 2.3.1.). Every tenth section, one was counterstained with thionine for cytoarchitectonic control.

### 2.3. Immunofluorescence (IMF)

#### 2.3.1. Choice of Sections from Series

Nissl-stained material was used to select sections, including all neocortical layers (layers 1–6), *cornu Ammonis* (CA) hippocampal fields and/or dentate gyrus, and the SVZ surrounding the lateral ventricles with the associated RMS and, when present, the OB. These sections spanned approximately a mediolateral extent of the brain from 100 µm to 900 µm lateral to the midline. For each series, eight equally-spaced immunostained sections (about one out of ten sections) were used for quantitative analysis.

#### 2.3.2. Single and Double IMF Procedures

We used single or double IMF routine procedures for light microscopy immunocytochemical detection. Sections were preincubated in 1% normal goat serum and 0.1% Triton X-100 for 1 h, and then incubated overnight in primary antibodies. Double-labeling experiments were performed by incubating sections in a mixture of two primary antibodies raised in different species, i.e., rabbit and mouse. After washing in 0.1 M phosphate-buffered saline pH 7.4, sections were incubated for 1 h with 1:500 anti-rabbit Alexa Fluor 594 (Thermo Fisher Scientific, Waltham, MA, USA, Cat# A11037) or 1:200 anti-mouse Alexa Fluor 488 (Thermo Fisher, Waltham, MA, USA, Cat# A11029) in single labeling experiments or a mixture of the secondary antibodies in double-labeling studies. After several washes in PBS, sections were mounted in fluorescence free medium (Cat# F6182, Fluoroshield, Sigma Chemicals, St. Louis, MO, USA) with or without nuclear counterstaining with 4′, 6-diamidino-2-phenylindole dihydrochloride (DAPI; Sigma Chemicals, St. Louis, MO, USA, Cat# D9564).

Sections were then photographed using a Leica DM6000 wide-field fluorescence microscope (Leica Microsystems, Wetzlar, Germany) with a 40× or a 63× lens using appropriate filter settings for each of the fluorophores used. Digital images were then merged using Photoshop CS6 (Adobe Systems, San Jose, CA, USA). Otherwise, sections were observed and photographed with a confocal microscope (Leica SP8, Leica Microsystems, Wetzlar, Germany).

### 2.4. Primary Antibodies and Controls

In this study, we used the following primary antibodies: 1:1500 polyclonal rabbit anti-γH2AX (Calbiochem, San Diego, CA, USA, Cat# DR1017); 1:200 monoclonal mouse anti-γH2AX (Abcam, Cambridge, UK, Cat# ab18311); 1:500 monoclonal mouse anti-γH2AX (Upstate Biotechnology, Lake Placid, NY, USA, Cat# 05-636); 1:1000 polyclonal rabbit anti-53BP1 (Abcam, Cambridge, UK, Cat# ab172580); 1:10 polyclonal rabbit anti-pHH3 (Abcam, Cambridge, UK, Cat# ab26127); 1:10 polyclonal rabbit anti-cCASP3 (Abcam, Cambridge, UK, Cat# 2302); 1:10 monoclonal mouse anti-BrdU (Bio-Rad Laboratories, Hercules, CA, USA, Cat# MCA2483). Using different anti-γH2AX primary antibodies raised in two distinct specieswas necessary to perform double-labeling IMF experiments with mixtures of primary or secondary antibodies. It also provided further validation of localization results. We evaluated differences in anti-γH2AX antibody reactiveness, as reported in Appendix A. Negative controls were performed by omitting the primary antibodies. As a result, the specific staining was completely abolished.

### 2.5. Quantitative Studies

#### 2.5.1. Counting γH2AX-Immunoreactive Nuclear Foci

Counting of γH2AX-immunoreactive nuclear foci was carried out in single labeled sections with the Calbiochem anti-γH2AX antibody using the Foci Counter program (Version 1, University of Konstanz, GE—http://focicounter.sourceforge.net/ accessed on 6 August 2021).

For each forebrain section, three areas were defined, according to current anatomical landmarks: the cerebral cortex, the hippocampus, and the SVZ/RMS/OB. In each of them, an observer that was unaware of the experimental group under scrutiny took five photographs using a 63× or 100× lens. Foci of all cells within each microscopic field were automatically counted with Foci Counter according to the software manual by adjusting the image threshold so that a clear identification of the nuclei in the “select” window was achieved (see Appendix B).

#### 2.5.2. Counting Immunoreactive Cells

Observers that were unaware of the experimental group under scrutiny directly counted single or double fluorescent cells and (when applicable) DAPI labeled nuclei in wide-field fluorescence photographs obtained with a 40× lens. For each forebrain section in the series, we selected and photographed at least five 320.43 µm × 239.34 µm microscopic fields of the cerebral cortex, hippocampus, and SVZ/RMS/OB. All immunoreactive cells as well as all DAPI fluorescent nuclei were counted in photographs. Manual counts in the hippocampus, namely in the dentate gyrus, where cell densities are very high, were further validated as described in Appendix C.

#### 2.5.3. Determination of Volumetric Cellular Densities

Volumetric cellular densities (cells/mm^3^) were calculated by counting the number of DAPI stained nuclei and immunostained cells in randomly chosen microscopic fields at 40× and multiplying the number of cells/mm^2^ per the actual thickness of sections. Details on the procedure are given in Appendix D.

### 2.6. Statistics

GraphPad Prism^®^ 9.0.2 (GraphPadSoftware, San Diego, CA, USA) was used for statistical analyses. These included descriptive statistics, normality tests to check for normally distributed data, and simple linear regression analysis.

Inferential statistics was carried out with parametric or nonparametric tests, according to need. Comparisons between two groups were carried out using either the unpaired two-tailed Mann–Whitney test or the unpaired t-test with Welch’s correction.

For comparisons of the three groups and normally distributed data, we used a conventional one-way ANOVA followed by the Tukey’s multiple comparison test, assuming equal standard deviations (SDs)—or the Brown-Forsythe and Welch ANOVA tests followed by Dunnett’s T3 multiple comparison test when SDs were unequal. For comparisons of the three groups with data that did not pass the D’Agostino and Pearson normality test, the non-parametric Kruskal–Wallis test, followed by Dunn’s multiple comparisons test, was employed.

An ordinary two-way ANOVA was carried out when it was necessary to analyze the statistical response and interaction of two different factors in determining the volumetric density or the percentage of γH2AX immunoreactive cells.

All data were reported as mean ± 95% confidence interval (CI). Values of *p* < 0.05 were considered statistically significant.

Venn diagrams were drawn using the software VennDiagramPlotter (Version 1, Pacific Northwest National Laboratory, Richland, WA, USA—https://omics.pnl.gov/software/venn-diagram-plotter accessed on 6 August 2021).

## 3. Results

### 3.1. X-ray Irradiation Induces a Strong γ Phosphorylation of H2AX and a Change in the Pattern of γH2AX Nuclear Staining

#### 3.1.1. γH2AX Immunoreactivity in Control Mice

Following our earlier study [13], we were able to detect some γH2AX immunoreactive nuclei, primarily in the cerebral cortexes, SVZ/RMS/OB, and hippocampi of control 24-month-old mice (Figure 2A,B). In these animals, γH2AX immunoreactive nuclei were relatively infrequent and mainly scattered throughout the gray matter. A qualitative map of their distribution was published in our previous study (see Figure 1 in [13]), where we also showed that most immunoreactive cells were neurons (although some glia were also stained).

Herein, we additionally counted the volumetric density (# cells/mm^3^) of γH2AX immunoreactive nuclei with two different primary antibodies (Abcam and Upstate) and observed no statistically significant differences with one or the other antibody in comparing each of the three forebrain areas under investigation separately (Figure 3A). More precisely, after Mann–Whitney test exact *p* values (Abcam vs. Upstate) were cerebral cortex = 0.1288, hippocampus = 0.3947, SVZ/RMS/OB = 0.3329.

We also calculated the percentage of γH2AX immunoreactive cells versus the total number of cells after nuclear staining with DAPI (Figure 4, black bars).

These data confirmed our previous observation [13] that H2AX was γ phosphorylated in the old mouse brain and demonstrated that γH2AX immunoreactive nuclei represent a small but consistent fraction (about 4–8%) of the total cell population throughout the forebrain.

#### 3.1.2. Effects of Irradiation on γ Phosphorylation of H2AX

In the two groups of irradiated mice (15 min and 30 min survival), γH2AX immunoreactivity occurred in a considerably higher number of cell nuclei than in the control group and was noticeably distributed in foci, whereas only rare cells displayed a diffuse pattern of nuclear fluorescence (Figure 2C,D).

Quantitative analysis showed that, following X-ray irradiation, the volumetric density (Figure 3B–D and Appendix E) and the percentage (Figure 4 and Appendix E) of γH2AX immunoreactive cells increased significantly, compared to controls, in all forebrain areas under study. Specifically, the percentage of immunoreactive cells increased from a minimum of 3.81 times in the cerebral cortex (Figure 4A) to a maximum of 13.69 times in the hippocampus after long survival (Figure 4B). Notably, the increase of γH2AX immunoreactive cells at any of the two-time points of survival was statistically significant compared to controls. However, there were no statistically significant differences in the cerebral cortex (Figure 4A) and hippocampus (Figure 4B) when the two groups of irradiated mice were compared, whereas a difference was evident in the SVZ/RMS/OB (Figure 4C).

These data demonstrate that H2AX is massively γ phosphorylated after X-rays in a considerable fraction of forebrain cells (up to 55% in the hippocampus). Phosphorylation of H2AX already occurs within 15 min from the start of irradiation and persists statistically unchanged in the cerebral cortex and hippocampus of long-surviving animals. On the other hand, in the SVZ/RMS/OB there is a further statistically significant increase between 15 min and 30 min survivors in the volumetric density (Figure 3D–yellow bars) and percentage (Figure 4C) of γH2AX immunoreactive cells (from about 31% to 38%) 30 min after irradiation.

The observation that in the SVZ/RMS/OB, γH2AX immunoreactivity increased further in long-surviving animals differently than in the cerebral cortex or hippocampus led us to hypothesize that the three brain areas had different levels of vulnerability to irradiation. To check for this, we carried out a two-way ANOVA of the volumetric densities of immunoreactive cells after staining with the Abcam and Upstate antibodies against γH2AX. Remarkably, under both conditions of immunostaining, the forebrain area was a statistically significant source of variation in the ANOVA. Results after immunostaining with the Abcam antibody were: Interaction, F = 17.40, *p* value < 0.0001; Forebrain areas, F = 90.07, *p* value < 0.0001; Irradiation, F = 58.853, *p* value < 0.0001. Results after immunostaining with the Upstate antibody were: Interaction F = 57.72, *p* value < 0.0001; Forebrain areas, F = 351.4, *p* value < 0.0001; Irradiation, F = 251.5, *p* value < 0.0001.

To confirm that the response to X-rays was indeed due to a DDR and to ascertain whether there was a relationship between the time of survival and the appearance of γH2AX immunoreactive nuclear foci, we calculated the number of foci/nuclei with the Foci Counter software (Figure 5). Although there were foci in γH2AX immunoreactive nuclei of control mice (see e.g., Figure 2A,B) their number and brightness were below the threshold in Foci Counter. Therefore, we limited our analyses to animals of the short and long survival groups.

Remarkably, the numbers of foci/nuclei were statistically different between 15- and 30 min survivors when the forebrain was considered (Figure 5A). Within the two groups of irradiated mice the number of foci/nuclei also differed among forebrain areas (Figure 5B,C) and when the results for the cerebral cortex, hippocampus, and SVR/RMS/OB were compared in short- and long-term survivors (Figure 5D–F).

Altogether, we concluded that, in response to irradiation, γ phosphorylation of H2AX increases over time, and there are statistically significant differences in the vulnerability to X-rays among forebrain areas.

### 3.2. Expression of 53BP1 in γH2AX Immunoreactive Cells after X-ray Irradiation

The well-known DDR component 53BP1 is recruited to the nucleus at the site of DNA damage, where it forms readily visualized foci [21,22,23]. Additionally, 53BP1 is an adaptor/mediator of DDR and is necessary for processing the DDR signal and recruitment of other repair factors. The recruitment of 53BPI to DNA DSBs requires γH2AX [22,24]. To confirm that phosphorylation of H2AX was indeed related to DDR, we have analyzed the expression and occurrence of 53BP1 immunoreactivity in combination with γH2AX in normal conditions and after X-ray irradiation (Figure 6, and Appendix F).

By direct microscopic inspection of γH2AX+53BP1 double-labeled preparations, we observed that irradiation, in parallel with the above-described raise in H2AX γ phosphorylation, also induced a very strong increment in the number of 53BP1 immunoreactive cells, most of which were also immunoreactive for γH2AX (Figure 6B,C). Remarkably, both labels mainly occurred in foci, rather than displaying a diffuse nuclear distribution. We then analyzed these preparations quantitatively and confirmed the results of microscopic observations (Figure 6D–G and Appendix F).

Altogether, these observations demonstrated that, after irradiation, γ phosphorylation of H2AX occurs together with a strong expression of 53BP1, thereby demonstrating the existence of a true DDR in irradiated cells.

### 3.3. An Increase of cCASP3 Labeled Cells Accompanies γ Phosphorylation of H2AX after X-ray Irradiation

Irradiation may lead to nerve cell death [25,26,27] and initiation of DNA fragmentation during apoptosis, inducing phosphorylation of H2AX [28,29]. Therefore, before addressing the intervention of CASP3 in the response to X-rays, we wanted to ascertain if there were measurable differences in the volumetric cellular density (measured after counting DAPI stained nuclei) among the three groups of experimental mice, i.e., if irradiation was sufficient to induce a massive cellular death within the brief temporal design of our study.

Kruskal–Wallis test, followed by Dunn’s multiple comparison test, demonstrated that differences between groups were not statistically significant. In the cerebral cortex, the results of Kruskal–Wallis test were approximate *p* value = 0.0524, Kruskal–Wallis statistics 5.899. Mean cellular densities were 150,865 ± 13,059 in control mice; 144,108 ± 21,583 in 15 min survivors; and 137,185 ± 12,069 in 30 min survivors (Dunn’s multiple comparison test: controls vs. 15 min survivors adjusted *p* value = 0.7172; controls vs. 30 min survivors adjusted *p* value = 0.1994; 15 min vs. 30 min survivors adjusted *p* value > 0.9999). In the hippocampus, the results of Kruskal–Wallis test were approximate *p* value = 0.9997, Kruskal-Wallis statistics 0.0005807. Mean cellular densities were 317,553 ± 35,217 in control mice; 324,701± 30,684 in 15 min survivors; and 316,557 ± 23,284 in 30 min survivors (Dunn’s multiple comparison test: controls vs. 15 min survivors adjusted *p* value >0.9999; controls vs. 30 min survivors adjusted *p* value > 0.9999; 15 min vs. 30 min survivors adjusted *p* value > 0.9999). In the SVZ/RMS/OB, the results of Kruskal–Wallis test were approximate *p* value = 0.8156, Kruskal-Wallis statistics 0.4077. Mean cellular densities were 104,495 ± 17,127 in control mice; 105,409 ± 14,425 in 15 min survivors; and 95,701 ± 7041 in 30 min survivors (Dunn’s multiple comparison test: controls vs. 15 min survivors adjusted *p* value > 0.9999; controls vs. 30 min survivors adjusted *p* value > 0.9999; 15 min vs. 30 min survivors adjusted *p* value > 0.9999). See Appendix D for further details.

These observations showed that, in the temporal window of our experiments, irradiation did not induce statistically detectable differences in the volumetric cell density of the forebrain areas under investigation. Yet they indicated a slight tendency toward reduction, particularly in the cerebral cortex.

With that established, we moved to analyze the distribution of cCASP3 immunoreactive cells in control, short survival, and long survival animals using single and double IMF procedures (Figure 7).

Microscopic observations showed that the number of cCASP3 immunoreactive cells tended to increase after irradiation, along with the number of γH2AX+cCASP3 double-labeled cells. Notably, cCASP3 immunoreactive cells displayed either a flawless condensed apoptotic morphology—i.e., they appeared in the classic form of apoptotic bodies (Figure 7C,G, K-arrows)—or a focal nuclear distribution of the fluorescence signal in nuclei that were intact (Figure 7K,O).

A statistical estimation of the volumetric cell density of cCASP3 immunoreactive cells confirmed the results of microscopic observations (Figure 8 and Appendix G). The increment in the number of cCASP3 immunoreactive cells started at 15 min survival in the cerebral cortex and SVZ/RMS/OB, where it further increased up to 30 min. Conversely, it only appeared at 30 min in the hippocampus, indicating a different behavior of this structure in the apoptotic response to irradiation.

We also calculated the linear regression curves of the changes in cCASP3 immunoreactivity versus the length of survival after X-ray irradiation (Figure 9). In all three forebrain regions, we observed that cCASP3 immunoreactive cells increased linearly in number but with different slopes (cerebral cortex: 18.83, *p* value = 0.0002; hippocampus: 16.65, *p* value = 0.0044; SVZ/RMS/OB: 87.48, *p* value < 0.0001). As is shown in the figure, the rate of increase was the highest in the SVZ/RMS/OB, whereas in the cerebral cortex and hippocampus rates were lower and very close to each other.

### 3.4. Colocalization of γH2AX and cCASP3 in Control and Irradiated Mice

Given the above-demonstrated increases in the number of γH2AX and cCASP3 immunoreactive cells after X-ray irradiation, and the occurrence of the two molecules in the same cells after microscopic observation (Figure 7), we calculated the percentage of co-localization of the two labels in each of the forebrain areas under investigation.

In control mice, the percentages of colocalization (cCASP3+γH2AX double-labeled cells/cCASP3 single-labeled cells) ranged from about 69% in the cerebral cortex and hippocampus to about 43% in SVZ/RMS/OB and did not display statistically significant differences between the three areas of the forebrain. The results of Kruskal–Wallis test followed by Dunn’s multiple comparisons were cerebral cortex vs. hippocampus adjusted *p* value > 0.9999; cerebral cortex vs. SVZ/RMS/OB adjusted *p* value = 0.4151; hippocampus vs. SVZ/RMS/OB adjusted *p* value = 0.4151.

These observations demonstrated that, under physiological conditions, there is a small number of cCASP3+γH2AX double-labeled cells in the forebrain, but these cells represent a sizable fraction of the total number of cCASP3 positive cells (see also Venn diagrams in the left column of Figure 10).

We then examined the effects of X-ray irradiation on γH2AX+cCASP3 co-expression. Notably, in the long survival group of mice, irradiation induced an increase in the volumetric density of double-labeled cells in all three forebrain areas (Figure 11; cerebral cortex: 2.2 times, hippocampus: 2.1 times, and SVZ/RMS/OB: 3.6 times). On the other hand, in short-surviving animals, the volumetric density of double-labeled cells was not statistically different in cerebral cortexes and hippocampi, but there was a noteworthy 1.9-fold increase in the SVZ/RMS/OB (Figure 11).

Venn diagrams of Figure 10 display the effects of irradiation on the percentages of double-labeled cells. Altogether, our observations demonstrated that irradiation induces a strong rise in the volumetric density of γH2AX immunoreactive cells and an apoptotic response. The latter was demonstrated by the increase in the volumetric density of cCASP3 immunoreactive cells that, for the most, were double-labeled, i.e., they also stained for γH2AX. They also suggested the existence of a different apoptotic response in SVZ/RMS/OB compared to cerebral cortexes or hippocampi.

We then reasoned that there were several possible explanations for the colocalization of γH2AX+cCASP3 in the same cells. The first was that CASP3 activation and H2AX γ phosphorylation arose stochastically and occasionally occurred together in the same cell but were unrelated. To test whether double-labeled nuclei could result from a random co-occurrence of two unconnected processes, we calculated the predicted percentage of co-labeling, i.e., the product of the observed percentages of every single label, and compared it with the observed percentage of γH2AX+cCASP3 double-labeled cells (Table 1).

From the Table, the observed percentages of co-labeling were much higher than the predicted random percentages of co-labeling in any of the experimental conditions tested.

It is noteworthy that, in all three forebrain areas of control mice, the difference between PCP and OPC was substantial. Such a result could be explained considering that in nonirradiated mice, the occurrence of DDR (and, hence, cCASP3 activation) was not simultaneously triggered by an external event. This results in very low percentages of cCASP3 immunoreactive cells (< %) that drastically reduce the value of PCP. Irrespective of the forebrain area and experimental group, co-labeling of cCASP3 and γH2AX occurred more frequently than predicted by chance, i.e., in case the expressions of γH2AX and cCASP3 are independent. Such an observation suggests that CASP3 activation and H2AX γ phosphorylation influence each other or that a common upstream process influences both.

### 3.5. An Increase in Cellular BrdU Incorporation Accompanies the H2AX Response to X-ray Irradiation

BrdU incorporation is extensively used to demonstrate DNA synthesis in living cells and has been widely employed to study adult neurogenesis in mammals, chiefly rodents (see [30] for discussion). On the other hand, cells with damaged DNA may attempt to de novo synthesize their genetic material during the DDR, and co-localization of DNA repair foci with BrdU foci indicates stalled or collapsed replication forks in these cells [31].

We thus analyzed the effects of X-ray irradiation on BrdU incorporation, and its colocalization with γH2AX. We observed that quite a few BrdU immunoreactive cells were present in the three forebrain areas of control mice and that irradiation led to a statistically significant increase in the volumetric density of positive cells (Figure 12). In control animals, the highest number of immunoreactive cells was observed in the SVZ/RMS/OB, followed by the hippocampus and the cerebral cortex. Irradiation led to an approximately six-fold increase in the cerebral cortex. The increase was fifteen-fold in the hippocampus and fourteen-fold in the SVZ/RMS/OB.

At microscopic observation, in both single- and double-labeling experiments, we saw that BrdU immunoreactivity was mainly punctate, with clear foci of positivity (Figure 13A,C,D,F; Figure 15A–C). Statistical analysis of BrdU+γH2AX immunostained preparations demonstrated that, in parallel with the increment of BrdU immunoreactive cells, irradiation also induced a rise in the volumetric total density of double-labeled cells (Figure 13G–J and Appendix J).

Venn diagrams in Figure 14 demonstrate the trend of γH2AX+BrdU co-labeling in control mice and after X-rays administration. Altogether, these series of results indicated that the DDR triggered by irradiation was accompanied by an attempt to synthesize new DNA, which, in all areas under study, was already evident after 15 min survival. Very likely, due to the limited number of BrdU positive cells, these data are subjected to quite large variations, which might explain the lack of statistical differences between short- and long-term survivors.

### 3.6. Increase in Cellular BrdU Incorporation after X-ray Irradiation Occurs Together with Phosphorylation of Histone H3

As we have shown in the previous sections, X-ray irradiation induces a DDR, i.e., the expression of γH2AX and 53BP1, accompanied by an increase of BrdU incorporation. Remarkably, however, H2AX can also be phosphorylated during mitosis [32], an event that may be completed before DNA repair occurs [33].

To gain more information on the relationship between BrdU incorporation and mitosis in response to X-rays, we analyzed the colocalization of BrdU and pHH3 [34] (Figure 15 and Appendix K).

Figure 15A–C show some exemplificative images of the pattern of BrdU+pHH3 colocalization in control and irradiated mice. Remarkably, in response to X-rays, both BrdU and pHH3 display immunoreactivity in foci. This pattern for pHH3 is typical of cells at the G_2_ and/or G_2_/M transition phase of their cycle [34]. We never observed pHH3 mitotic chromosomes. (Figure 15D–G) demonstrate the results of X-ray exposure onto the volumetric density of BrdU+pHH3 immunoreactive cells after statistical analysis. From their inspection, it appears that irradiation not only led to increased incorporation of BrdU as above shown (Figure 12 and Figure 13), but also to augmented numbers of BrdU+pHH3 double-labeled cells in the cerebral cortex (Figure 15E) and SVZ/RMS/OB (Figure 15G), but not hippocampus (Figure 15F).

Venn diagrams in Figure 16 show the trend of BrdU+pHH3 co-labeling in control mice and after X-rays administration. If we consider the forebrain, it is remarkable that, in control mice, BrdU+pHH3 double-labeled cells represented 5.55% of the total number of BrdU-only immunoreactive cells, but such a percentage increased substantially in those animals that survived 15 or 30 min—to 24.41% and 25.64%, respectively. Altogether, our results indicated that an attempt to synthesize new DNA and a mitotic/pseudo-mitotic response in the damaged cells accompanied the DDR triggered by irradiation.

## 4. Discussion

We demonstrated that irradiation not only induced a robust DDR with an intense phosphorylation of H2AX and upregulation of 53BP1 but that DDR was accompanied by an attempt to synthesize new DNA—evident from the rise in BrdU immunostained cells—and increased expression of cCASP3. Notably, the association of γH2AX, 53BP1, and cCASP3 was also evident in normal nonirradiated mice, where DNA synthesis was likely related to the disturbances in DNA repair mechanisms, rather than to a bona fide cell division, as inferred from BrdU + pHH3 double-staining experiments.

Although phosphorylation of H2AX is an initial step in DDR, the histone has also been implicated in cell proliferation and apoptosis (see Introduction), and, very recently, in the support of neuronal maintenance via controlling mitochondrial homeostasis [35]. Given the pleiotropic effects of the molecule—and considering that the links between cell proliferation and apoptosis in adult and old neurogenesis are quite complex [15]—we have split our discussion to first consider the γH2AX/cCASP3 response to irradiation, a well-established DNA damaging event, and then the presence of the two molecules in the normal aging brain.

### 4.1. γH2AX/cCASP3 Response to X-ray Irradiation

We demonstrated that whole-brain X-ray irradiation with a 10 Gray dose induces strong phosphorylation of H2AX with the appearance of nuclear foci of IMF that are revealing of the occurrence of a DDR in irradiated cells [19]. Irradiated mice were left to survive for 15 or 30 min as phosphorylation of H2AX reached a peak within this interval, to decline thereafter [19,20].

Studies on yeast strains aiming to analyze the temporal trend of H2AX phosphorylation after ionizing radiations have demonstrated that, after a very rapid rise and a short plateau of approximately one hour, the number of γH2AX foci started to decrease two hours post-irradiation and returned to baseline within six hours [20]. The results of our observations in mice that survived 15 or 30 min following irradiation (Figure 1 and Figure 2) are fully in line with these studies on unicellular organisms. After calculating the volumetric density of γH2AX immunoreactive cells (Figure 3), we notably saw that, except for the SVZ/RMS/OB, where immunoreactive cells/mm^3^ increased further at 30 min survival, there was not a statistically significant increase between short- and long-term survivors, confirming that we were essentially monitoring the plateau of the H2AX response to X-rays. The trend was the same when the percentage of γH2AX immunoreactive cells was considered versus the total number of nuclei after staining with DAPI (Figure 4). However, DNA damage was in progress between 15 and 30 min, as the number of γH2AX foci/nucleus was higher in long-survivors (Figure 5). Such an interpretation is in agreement with the notion that the number of foci/nuclei after irradiation, as well as the amount of γH2AX, is directly dependent on the number of DSBs and, thus, the extent of DNA damage [19]. The occurrence of a DDR in response to irradiation was confirmed in γH2AX+53BP1 double-labeling experiments, which showed statistically significant increases in the volumetric density of double-labeled cells after X-rays (Figure 6) following the notion that 53BP1 is an established DDR component [21,22,23].

Having confirmed a direct relationship between DNA damage and H2AX γ phosphorylation in response to X-rays, we wanted to investigate the fate of the cells with damaged DNA.

An early (1998) in vitro study on rat cortical neurons and astrocytes postulated that DNA damage could cause neuronal apoptosis, that the amount of damage could determine the degree of apoptosis induced, that slow repair of damage could play a role in the susceptibility of neurons to apoptosis, and that astrocytes were relatively resistant to death [25]. In that study, apoptosis was assessed by observing DNA fragmentation in Southern blots, and thus, no information was gathered on the possible intervention of CASP3, whose role in apoptosis was to be discovered more precisely in those years [36,37]. Subsequently, the same group demonstrated that neuronal death was blocked in vitro by a pan-caspase inhibitor when cells were irradiated with moderate X-ray doses (≤32 Gray), whereas cell death was unresponsive to caspase inhibition after severe DNA damage (up to 128 Gray) and could occur by non-apoptotic (necrotic) mechanisms [26]. Very recently, similar results were obtained by another group that studied (in vitro and in vivo) the inhibition of DNA repair pathways in mouse cortical neurons following whole-body irradiation at 10 Grays [27].

Nonetheless, the temporal and causative relationship between γH2AX and cCASP3 remained somewhat elusive. Further studies in cultured cell lines showed that the formation of γH2AX was an early chromatin modification following the commencement of DNA fragmentation during apoptosis [28]. In addition, evidence has been published showing that, in aging and diseases, such as Alzheimer’s disease, unhealthy neurons aberrantly re-enter into the cell cycle, with lethal consequences leading to an apoptotic demise [29].

To shed more light on such a relationship, we carried out a series of immunocytochemical cCASP3+γH2AX double-labeling experiments after X-ray irradiation. These experiments demonstrated that irradiation increased the number of double-labeled cells. Notably, the observed percentages of colocalization between the two labels after X-rays were higher, compared to those predicted from the simple product of the observed percentages of every single label (Table 1). This excluded the possibility that the co-localization of γH2AX and cCASP3 at the single-cell level was the result of a stochastic process. This in turn led to the possibility that H2AX γ phosphorylation primes CASP3 activation or vice versa, or that an upstream process is responsible for the activation of both molecules. Given the experimental layout here employed, we consider it reasonable to suggest that X-ray irradiation is the upstream process accountable for the increase in the volumetric density of γH2AX+cCASP3 immunoreactive cells and the percentage of colocalization of the two labels. In addition, we believe it odd to hypothesize that activation of CASP3 precedes H2AX γ phosphorylation for two reasons: first, cCASP3-immunoreactive cells are only a small fraction of the cells immunoreactive for γH2AX; second, the time course of the number/mm^3^ of these cells is somewhat directly proportional to the interval of survival after X-rays.

Therefore, irradiation leads to a DDR with rapid phosphorylation of H2AX, expression of 53BP1, and activation of CASP3. The latter event was slower than the rise of γH2AX, very likely as a consequence of the moderate (10 Gray) [26] X-ray dosage employed. Therefore, cCASP3 was detected in a small percentage of γH2AX immunoreactive cells, and activation of the caspase did not result in a measurable reduction of the volumetric cellular density in any of the forebrain areas under study (Appendix D, Figure A4). Although to the best of our knowledge there have been no direct observations on the relationship between H2AX phosphorylation and CASP3 activation in the intact aging brain, previous studies have shown that ionizing radiations induce DNA damage and may initiate neuronal apoptosis [25,26,38,39] with activation of CASP3 [27,40].

Prior experiments have shown that γH2AX not only formed at the onset of the DDR but also in the initial phases of apoptosis induced by death receptor activation [28]. Subsequent microscopic studies revealed a particular pattern of immunostaining when H2AX was phosphorylated because of apoptosis [6]. In this case, the pattern of apoptotic γH2AX immunostaining was different from the well-established DDR focal appearance, as H2AX phosphorylation initiates at the nuclear periphery immediately inside the nuclear envelope, while total H2AX remains distributed throughout the nucleus. This process was readily detectable by IMF and has been referred to as the γH2AX ring, which coincides with the cellular site of localization of another phosphorylated histone, the H2B (see also [6]). Remarkably—and in keeping with these reported observations in normal and cancer cells in vitro—we have been unable to obtain evidence of the presence of the apoptotic γH2AX ring, nor of the onset of phosphorylated H2B (data not shown) in any of our experimental conditions. Thus, our experiments are in full accordance with the belief that activation of CASP3 is a secondary event to DDR and the onset of γH2AX is not a primary consequence of apoptosis (see also Figure 5 in [6]).

Another point of discussion regards the effects of irradiation on the incorporation of BrdU and the detection of pHH3 in our material. Although often (mis)used to identify proliferating cells, particularly in the study of mammalian (chiefly rodents and monkeys) adult neurogenesis [30,41], BrdU is truly an S-phase marker. Contrarily, pHH3 is a bona fide marker of dividing cells displaying a nuclear distribution in foci or labeling mitotic chromosomes in the G_2_ and M phases of the cell cycle, respectively [42,43]. In the present work, we demonstrated that irradiation induced an increase in the volumetric density of BrdU immunoreactive cells (Figure 12), in parallel with a statistically significant rise of BrdU + γH2AX double-labeled cells (Figure 13). Therefore, the incorporation of BrdU in the forebrain cells of irradiated mice should be correctly interpreted as an attempt to respond to the DNA damage caused by X-rays with a de novo synthesis of DNA. Alternatively, or in addition to, incorporation of BrdU can be the first step of an aberrant mitotic response. Such a possibility is consistent with the results of our BrdU + pHH3 double-labeling experiments, in which we observed an increased volumetric density of BrdU + pHH3 immunoreactive cells after X-rays (Figure 15), and a rise in the fraction of BrdU immunoreactive cells that also stained for pHH3 (Figure 16). Interestingly, in vitro studies aiming to compare the genotoxic potential of different chemicals have employed benchmark dose analyses of γH2AX and pHH3 endpoints for quantitative comparison [44,45]. Genotoxicity may be consequent to the effects of clastogens and aneugens. The former act directly on the DNA and cause structural aberrations of chromosomes, which potentially trigger mutations via repairing errors, whereas the latter are DNA-nonreactive and interfere with the mitotic process. Remarkably, clastogens raise H2AX, whereas aneugens, such as spindle poisons, increase pHH3 [45]. As the genotoxic mechanism of ionizing radiations is clastogenic, and co-localization of DNA repair foci (γH2AX) with BrdU foci indicates stalled or collapsed replication forks [31], we believe it reasonable that expression of pHH3, together with BrdU in our experimental context, derives from an aberrant mitotic response or a mitotic catastrophe, as demonstrated in different species and pathological conditions [46,47,48,49].

Finally, we would like to comment briefly about the potential translational significance of our results. Patients undergoing whole-brain radiation therapy suffer from a series of side effects, including severe and irreversible cognitive decline [50,51]. Remarkably, the radiation-induced neurocognitive impairment is classically thought to be the consequence of multiple mechanisms (i.e., neuroinflammation, decreased neurogenesis, reduced proliferation of vascular and glial cells), chiefly (but not only) affecting the hippocampus [52].

Using a mouse model and γH2AX immunocytochemistry, it was recently suggested that early life irradiation induced persistent DNA damage foci at later stages of lifetime, resulting in more rapid structural and cognitive aging and shortened life expectancy [53]. Our results indicated that an additional mechanism of cognitive impairment in the aging brain may be represented by an early-onset increased susceptibility to apoptosis, which leads to widespread cell death in the neocortex and hippocampus, two structures deeply involved in cognitive functions. Death could affect not only the neurons [38] but also the glia and cells of the small blood vessels, which can proliferate and are responsible for the vascular cognitive impairment associated with subcortical small vessel disease [54]. This view is also supported by the efficacy of antiapoptotic drugs, such as memantine, for the prevention of radiation-induced cognitive decline [55]. Thus, therapeutic interventions to reduce the cognitive impact of radiation therapy might be supported by additional strategies in the future.

### 4.2. γH2AX/cCASP3 in the Normal Aging Brain

As mentioned, we investigated the effects of X-ray irradiation on the aged mouse brain to better understand the significance of H2AX phosphorylation and associated events under normal conditions. We previously demonstrated that γH2AX was detectable in the brains of old mice that were not subjected to any known DNA injury [13]. Given that—as already discussed—H2AX phosphorylation could also be a consequence of apoptosis, our previous data were not conclusive about the existence of a true DDR in these mice.

Herein, we have demonstrated that the early components of the DDR are present in small but sizable populations of forebrain cells in untreated mice. Cells immunoreactive for γH2AX could be detected in control animals using two different primary antibodies against the phosphorylated form of the histone (Figure 2A,B, Figure 3 and Figure 4). Quantitatively, they were about 5000 to 18,000/mm^3^ depending on forebrain areas and antibodies under use (Figure 3A) and around 4–8% of the total cell populations of the forebrain areas under study, as estimated after nuclear staining with DAPI (Figure 4). Remarkably, results for the volumetric density of γH2AX+53BP1 double-stained cells (from about 4000 to 8500 cells/mm^3^) showed that, as after X-ray irradiation, a large fraction of γH2AX immunoreactive cells also expressed 53BP1 (Figure 6A,D–G). Thus, these cells were undergoing a bona fide DDR, in keeping with the demonstration of persistent DNA damage in senescent cells and aged mammalian tissues [56]. In addition, our observations are in full agreement with the current notion that H2AX acts as a guardian of the genome integrity by operating on poorly repaired DSBs that increase the mutation burden of the cells, conceivably leading to genome instability and carcinogenesis [57].

In untreated aging mice, the pattern of immunostaining and the quantitative estimations for cCASP3, BrdU, and pHH3 were analogous to those observed after irradiation, except that they regarded consistently smaller numbers of cells (Figure 7A–D, Figure 8, Figure 10 and Figure 11). In addition, in these mice, there were clear differences among forebrain areas (see below). Quite likely, these differences were related to the nonsynchronous occurrences of the DNA damaging events, differently from the experimental induction by ionizing radiations.

We do not know the exact source and type of DNA damage in untreated mice. However, it would be interesting to speculate on the intervention of retrotransposons in the generation of the DSBs that may trigger the phosphorylation of H2AX in normal aging. Retrotransposons are mobile DNA elements that can change their position within the genome, producing genomic diversity between neurons and intervening in certain neurological disorders [58]. Remarkably, the human LINE-1 retrotransposon generates high levels of DNA DSBs in HeLa cells, as detected with γH2AX immunocytochemistry and single-cell gel electrophoresis [59].

Regardless of the cause of DNA DSBs, the simultaneous cellular expression of γH2AX and 53BP1 in aged animals is coherent with the possibility that, at low-to-mid natural doses of DNA damage, the H2AX-mediated concentration of 53BP1 at DSBs is crucial to trigger a DDR. The latter would then directly result in apoptosis of postmitotic neurons or prevent the entry of damaged proliferating cells into mitosis [21,24].

In addition, expression of 53BP1 and cCASP3 in the aged mouse brain can be linked to the occurrence of SSBs, because, as mentioned in the Introduction, the accumulation of p53 in SSBs is faster than in DSBs, and cells with SSBs undergo apoptosis more easily [4].

Finally, apoptosis-associated DNA fragmentation may be an extra source of DSBs in cell chromatin [2], in keeping with the observed association of γH2AX and cCASP3.

### 4.3. Is There a Different Vulnerability to DNA Damage among Forebrain Areas Related to a Diverse Differentiation State of Neurons?

Our observations suggest that there may be differences in the vulnerability of the cerebral cortex, hippocampus, and SVZ/RMS/OB to DNA damage. The response to X-rays, for instance, displayed different behaviors in the SVZ/RMS/OB in comparison to the two other areas of the forebrain Namely, in the latter, γH2AX immunoreactive cells/mm^3^ (Figure 3D), their percentage versus the total number of DAPI-stained cells (Figure 4C), and the volumetric density of γH2AX+cCASP3 immunoreactive cells (Figure 11C) continued to increase in 30 min survivors, differently from the cerebral cortex and hippocampus.

We have previously demonstrated that most γH2AX immunoreactive cells in the untreated old mouse brain were neurons expressing the neuronal nuclear antigen (NeuN), though some glial fibrillary acidic protein (GFAP) immunostained glia was also stained [13]. Both types of cells displayed interkinetic nuclei with a few foci of γH2AX positivity, as observed in this study. In terms of their differentiation state, adult/aging neurons in the cerebral cortexes, hippocampi, and SVZ/RMS/OB display remarkable differences. The cerebral cortex is, in large part, if not exclusively, populated by postmitotic nondividing neurons, whereas the hippocampal dentate gyrus and the SVZ/RMS/OB are sites of adult neurogenesis, and it has been suggested that some neurons could deviate from the RMS to reach other brain regions among which the prefrontal cortex [60]. For these reasons, we have briefly discussed, herein, the cerebral cortex from one side and the hippocampus and SVZ/RMS/OB from the other.

#### 4.3.1. Cerebral Cortex

In our initial paper [13], we used single or multiple BrdU injections to label DNA-synthesizing cells in the cerebral cortex. Sections were not subjected to a systematic random sampling, as done here, but sampled along the rostrocaudal axis of the brain at eight specifically defined distances from bregma (see Figure 6 in [13]). The mean value of γH2AX+BrdU colocalization was 59% (±2.44) after a single BrdU administration. Such a figure was close to that calculated here (60% ± 14.5), even though the brain was, in this case, sectioned along the transversal axis. What we did not know at that time was whether BrdU incorporation was linked to proliferation and/or apoptotic events. We here provide statistically significant proof that in control not-irradiated mice there is no colocalization of BrdU and pHH3 (Figure 15E and Figure 16) in the presence of fractions of γH2AX immunoreactive cells that also express BrdU (Figure 13H and Figure 14) or cCASP3 (Figure 10 and Figure 11A). Therefore, we can now safely conclude that BrdU incorporation in the old cerebral cortex is part of a naturally occurring DDR with phosphorylation of H2AX, recruitment of 53BPI, and activation of CASP3, eventually leading to death, in agreement with the known intervention of γH2AX in the modulation of checkpoint responses [61].

Therefore, our present study reveals that the aged mouse cerebral cortex is prone to DNA damage. The volumetric density of cCASP3 immunoreactive cells is higher than that of BrdU positive cells (compare Figure 11A and Figure 12A). Thus, it may be possible that at least a fraction of the cCASP3 positive cells undergo a rapid apoptotic death having accumulated too many unrepaired SSBs [4]. In parallel, it seems reasonable that other cells respond to DSBs with an initial attempt to repair their DNA (and hence incorporate BrdU), which is eventually followed by CASP3-dependent apoptosis. It is also conceivable that a pseudo-DDR, i.e., the appearance of γH2AX foci in the absence of detectable DNA breaks, also occurs in the aging mouse cortex, as demonstrated in a senescent fibroblast cell line [62]. However, such an event seems highly improbable, as there was no accumulation of 53BP1 in γH2AX foci in senescent fibroblasts [62], despite what we have observed in cortical cells.

DNA repair systems should be of great importance in the adult and old brain [63,64,65], and efficient DNA repair is surely needed in long-living postmitotic cells with no or limited regeneration from precursors. However, a further point of conjecture regards the possibility that γH2AX immunoreactive neurons in the old cerebral cortex are senescent-like neurons surviving with a persistently activated DDR [29,66]. These senescent-like cells may be resistant to apoptosis and prone to inflammation and neurological dysfunction [29]. Notably, the operational effects of an age-dependent initiation of the DDR with the appearance of senescent-like neurons may be very important for neurodegeneration, cognitive decline, and dementia [67,68].

Finally, the cerebral cortex can be partially included in the list of the novel neurogenic zones in the adult brain, possibly as the site of migration of new neurons from the SVZ/RMS or as a consequence of the proliferation of local progenitors [60]. The discussion about the adult neurogenic potential of the cerebral cortex is beyond the scope of this paper. However, also in light of our current observations, it seems reasonable to suggest prudence in interpreting the results of studies that have been often carried out in young adult animals (around P60 for mice) or pathological material. These studies were frequently based exclusively on BrdU incorporation to claim for the occurrence of proliferation (see Table 3 in [60]). Indeed, it seems possible that BrdU positive cells at this age are simply the remnants of rare proliferating cells that will disappear soon thereafter.

#### 4.3.2. Hippocampus and SVZ/MS/OB

The interpretation of our present findings on the phosphorylation of H2AX in the hippocampus and SVZ/RMS/OB is made more complex by the existence of adult neurogenesis in these two brain niches [60]. Our prior semi-quantitative study [13] indicated that the two neurogenic niches display a different behavior concerning H2AX phosphorylation and left open the question of whether activation of H2AX was related to the occurrence of DNA DSBs during cell division, apoptosis, or simply to physiological activity, as demonstrated by others in hippocampi [69].

The following additional information was gathered from the present results:Phosphorylation of H2AX is accompanied by an expression of 53BP1 in both hippocampus and SVZ/RMS/OB (Figure 6F,G) as proof that it is linked to DDR.Both areas contain a fraction of cCASP3 immunoreactive cells (Figure 7C,K,O and Figure 8B,C) some of which co-express γH2AX (Figure 7A–D,I–P, Figure 10 and Figure 11B,C) to indicate that, at least, these cells are committed to apoptosis.The hippocampus, as well as the SVZ/RMS/OB, includes a small population of BrdU immunoreactive cells (Figure 12B,C) that, in part, additionally express γH2AX (Figure 13I,J and Figure 14).A subpopulation of BrdU immunoreactive cells express pHH3 in the hippocampus but not SVZ/RMS/OB (Figure 15F,G and Figure 16), suggesting that phosphorylation of H2AX in BrdU positive SVZ/RMS/OB cells may not be necessarily linked to true proliferation but rather to the pseudo-proliferative state that is observed in dividing cells entering a senescent state [29].

Interestingly, an increase in DSBs (measured with γH2AX immunocytochemistry) was observed in the six-month-old mouse hippocampi, where neuronal activity was experimentally increased by sensory or optogenetic stimulation and exacerbated by amyloid β with an aggravation of synaptic dysfunctions [69]. Thus, although undoubtedly further investigations are needed, it seems reasonable to infer that aged hippocampal cells expressing the DDR markers γH2AX and 53BP1, as well as activated CASP3, have accumulated an amount of DSBs that is incompatible with survival. Consequently, incorporation of BrdU in these cells may be linked to an aberrant re-entry into the cell cycle before undergoing apoptosis [29].

Similar events could take place in the SVZ/RMS/OB, where γH2AX was detectable starting from embryonic life [13]. In the mouse brain, the SVZ is a neurogenic stem cell niche [60] that persists throughout life, albeit progressively reducing in old age [70,71,72]. Numerous studies have utilized (and often misemployed) the S phase marker BrdU to detect proliferating cells in this forebrain area [30]. The occurrence of BrdU + γH2AX immunostained cells and the absence of BrdU + pHH3 double-stained cells indicates that the cells expressing γH2AX in the SVZ/RMS/OB system are in the S phase of their cycle. Therefore, BrdU incorporation may, in these cells, be related to a DDR during abortive mitosis [32], eventually leading to CASP3-dependent apoptosis. In full agreement with this interpretation of our findings, a reduction of BrdU incorporation in the SVZ of old mice was previously reported, and it was conjectured that enhanced apoptosis in aged animals could be responsible for lowering the number of S phase cells [72]. Additionally, when the generation of new neurons was monitored over time after BrdU incorporation, up to 70% reduction in the number of progenitors and young neurons was detected over several months [73,74], indicating that apoptosis is a common element during adult neurogenesis [75] and sharply prevails over proliferation in the old brain, as we hypothesized.

## 5. Conclusions

We have taken advantage of the well-known capacity of X-rays to induce DSBs in the cell DNA and a consequent DDR to demonstrate that:γH2AX associates with 53BP1 after irradiation and is thus a reliable marker of DDR in vivo.Many γH2AX immunoreactive irradiated cells undergo apoptosis with cleavage of CASP3 after failing to repair their genetic material, notwithstanding the incorporation of BrdU as an indicator of de novo DNA synthesis.There is a close association of γH2AX, 53BP1, and cCASP3 in untreated mice, indicating that cells in the cerebral cortex, hippocampus, and SVZ/RMS/OB are subjected to some sort of naturally occurring endogenous DNA damage, directly in the form of DSBs or as unrepaired SSBs that are then converted to DSBs.The incorporation of BrdU—with or without the concurrent expression of pHH3—may be related to the direct activation of a DDR (in cortical postmitotic neurons) or an aberrant cell cycle reentry with possible evolution into a senescent-like state (in the neurogenic hippocampus and SVZ/RMS/OB), respectively.

## Figures and Tables

**Figure 1 biomedicines-09-01166-f001:**
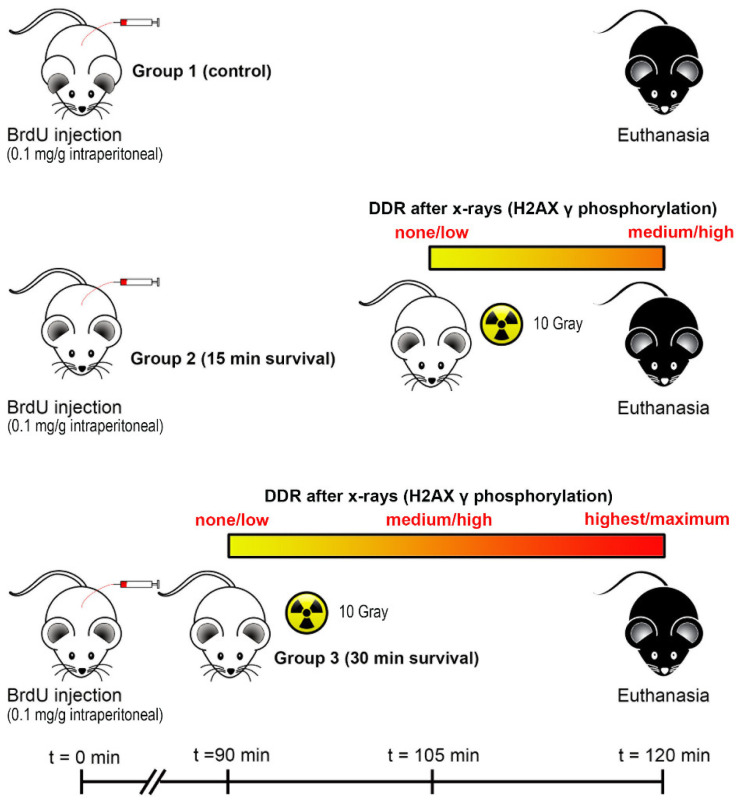
Schematic layout of the X-ray irradiation and control experiments. Experiments are designed such that all animals survive exactly two hours after injection of the DNA synthesis marker BrdU. The colored bars represent, pictorially, H2AX γ phosphorylation, according to literature data showing that γH2AX appears rapidly following cell exposure to ionizing radiations [18]. The amount of γH2AX directly correlates to the number of DSBs and, thus, to the extent of DNA damage [19]. As such, this experimental layout allows the study of the early temporal evolution of H2AX γ phosphorylation/DNA damage in vivo [20].

**Figure 2 biomedicines-09-01166-f002:**
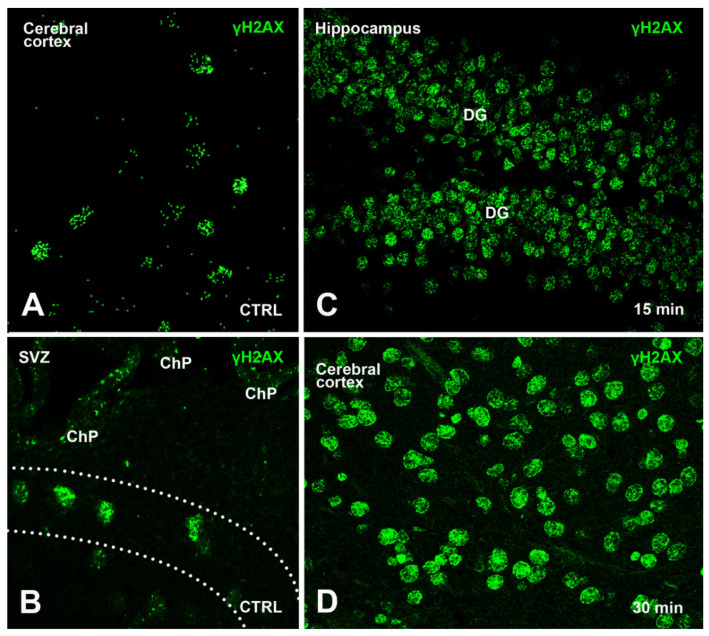
Exemplificative confocal images of γH2AX immunoreactive cells in control and irradiated mice after immunostaining with the Upstate anti-γH2AX antibody. (**A**) Nuclear distribution of γH2AX immunoreactivity in the cerebral cortex of a control mouse. (**B**) γH2AX immunoreactive cells in the SVZ of a control mouse. The dashed lines mark the limits of the SVZ. (**C**) γH2AX immunoreactive cells in the hippocampal dentate gyrus after 15 min irradiation. (**D**) Increase in the number of γH2AX immunoreactive nuclei in the cerebral cortex after 30 min irradiation. Compare with panel A and note the appearance of numerous foci of immunoreactivity. Abbreviations: CTRL = control; ChP = Choroid plexus of lateral ventricle; DG = dentate gyrus of hippocampus. Original magnifications 40×.

**Figure 3 biomedicines-09-01166-f003:**
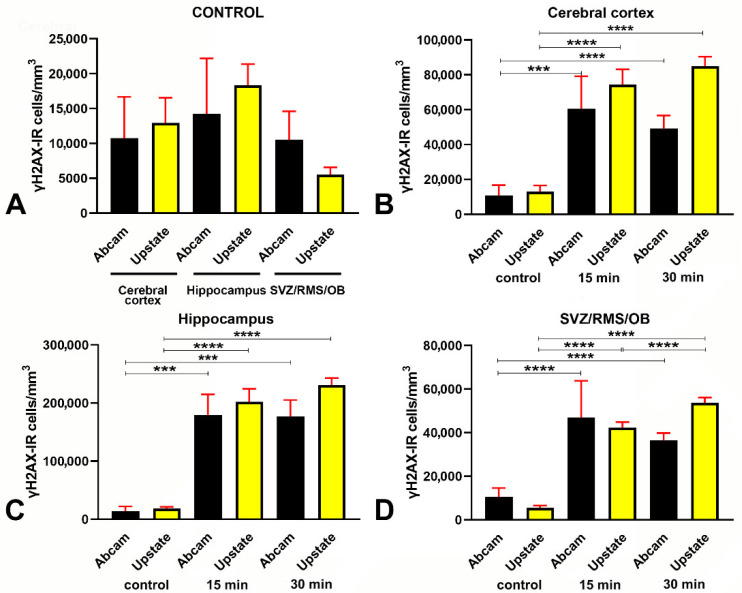
Volumetric density of γH2AX immunoreactive cells in control and irradiated mice after immunostaining with two different polyclonal rabbit antibodies. (**A**) Lack of statistically significant differences in the number of γH2AX immunoreactive cells/mm^3^ after immunostaining with the Abcam or the Upstate primary antibody in control (nonirradiated) mice. (**B**–**D**) Statistically significant increase in the number of γH2AX immunoreactive cells/mm^3^ in the cerebral cortex (**B**), hippocampus (**C**), and SVZ/RMS/OB (**D**) following X-ray irradiation. Trends are fully comparable after immunostaining with both primary antibodies. Data are expressed as mean ± 95% CI. *** 0.001 > *p* ≥ 0.0001, **** *p* < 0.0001. Bars are 95% CI. Numbers of statistical data are reported in Appendix E.

**Figure 4 biomedicines-09-01166-f004:**
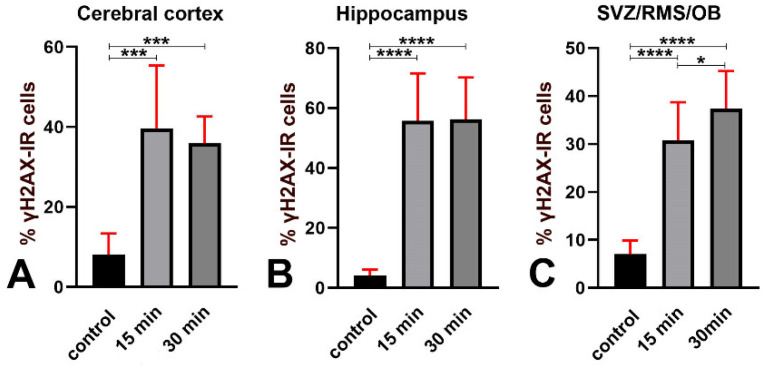
Percentages of γH2AX immunoreactive nuclei versus the total number of nuclei after labeling with DAPI in the cerebral cortex (**A**), hippocampus (**B**), and SVZ/RMS/OB (**C**) in control and irradiated mice. * 0.05 > *p* ≥ 0.01, *** 0.001 > *p* ≥ 0.0001, **** *p* < 0.0001. Bars are 95% CI. Immunostaining was performed with the Abcam antibody. Numbers of statistical data are reported in Appendix E.

**Figure 5 biomedicines-09-01166-f005:**
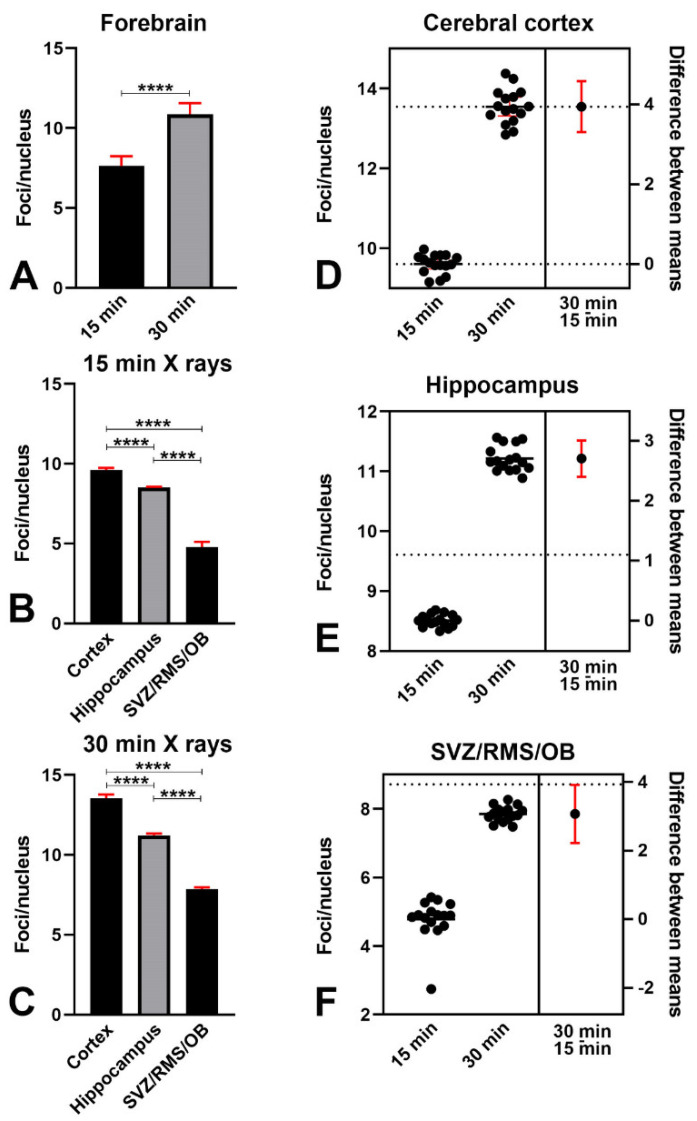
Number of γH2AX immunoreactive foci per nucleus in short- and long-term survivors after X-ray irradiation. (**A**): Difference in the mean number of immunoreactive foci/nuclei in the forebrain at 15- and 30 min survival (15 min: 7.633 ± 0.616; 30 min: 10.87 ± 0.69; Mann–Whitney test, two-tailed exact *p* value < 0.0001). (**B**) Difference in the mean number of foci/nuclei among forebrain areas in short survival (15 min) group. Ordinary one-way ANOVA followed by Tukey’s multiple comparisons test (cerebral cortex: 9.609 ± 0.129; hippocampus: 8.510 ± 0.054; OB/SVZ/RMS 4.779 ± 0.3275–adjusted *p* values for all comparisons < 0.0001). (**C**) Difference in the mean number of foci/nuclei among forebrain areas in long-term survival (30 min) group. Ordinary one-way ANOVA followed by Tukey’s multiple comparisons test (cerebral cortex: 13.55 ± 0.23; hippocampus 11.21 ± 0.12; OB/SVZ/RMS 7.852 ± 0.12; adjusted *p* values for all comparisons < 0.0001). (**D**) Differences in the mean number of foci/nuclei in the cerebral cortex after short- and long-term survival. Unpaired t-test with Welch’s correlation; two-tailed *p* value < 0.0001. (**E**) Differences in the mean number of foci/nuclei in the hippocampus after short- and long-term survival. Unpaired t-test with Welch’s correlation–two-tailed *p* value < 0.0001. (**F**) Differences in the mean number of foci/nuclei in the SVZ/RMS/OB after short and long survival. Unpaired t-test with Welch’s correlation; two-tailed *p* value < 0.0001. **** *p* < 0.0001. Bars are 95% CI. Staining was carried out with the Calbiochem primary antibody against γH2AX.

**Figure 6 biomedicines-09-01166-f006:**
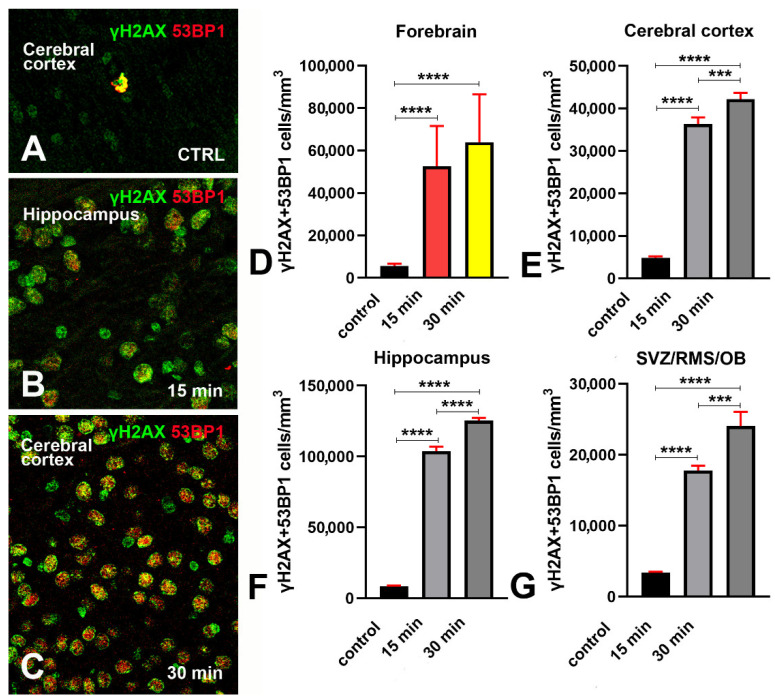
γH2AX + 53BP1 double-labeling experiments in the forebrain of control and irradiated mice. (**A**) exemplificative image of the pattern of immunostaining in the cerebral cortex of a control nonirradiated mouse. Note the double-labeled cell displaying immunoreactive foci singularly labeled for γH2AX (green) or 53BP1 (red) and foci of dual immunoreactivity (yellow). (**B**) exemplificative image of the pattern of immunostaining in the hippocampus of a mouse that survived 15 min after irradiation. Note the high number of immunoreactive cells, some of which display distinct nuclear foci of γH2AX (green) or 53BP1 (red) immunoreactivity. (**C**) exemplificative image of the pattern of immunostaining in the cerebral cortex of a mouse that survived 30 min after irradiation. Note that most of the cells are double-labeled and display numerous yellow foci of immunoreactivity indicative for γH2AX (green) + 53BP1 colocalization (red). Original magnifications 40×. (**D**–**G**) Quantitative analysis of the volumetric density of double-labeled cells, in the whole forebrain, cerebral cortex, hippocampus, and SVZ/RMS/OB. Data are expressed as mean ± 95% CI. *** 0.001 > *p* ≥ 0.0001, **** *p* < 0.0001. Bars are 95% CI. Staining for γH2AX was performed using the Upstate primary antibody. For numerical data of statistics see Table A2.

**Figure 7 biomedicines-09-01166-f007:**
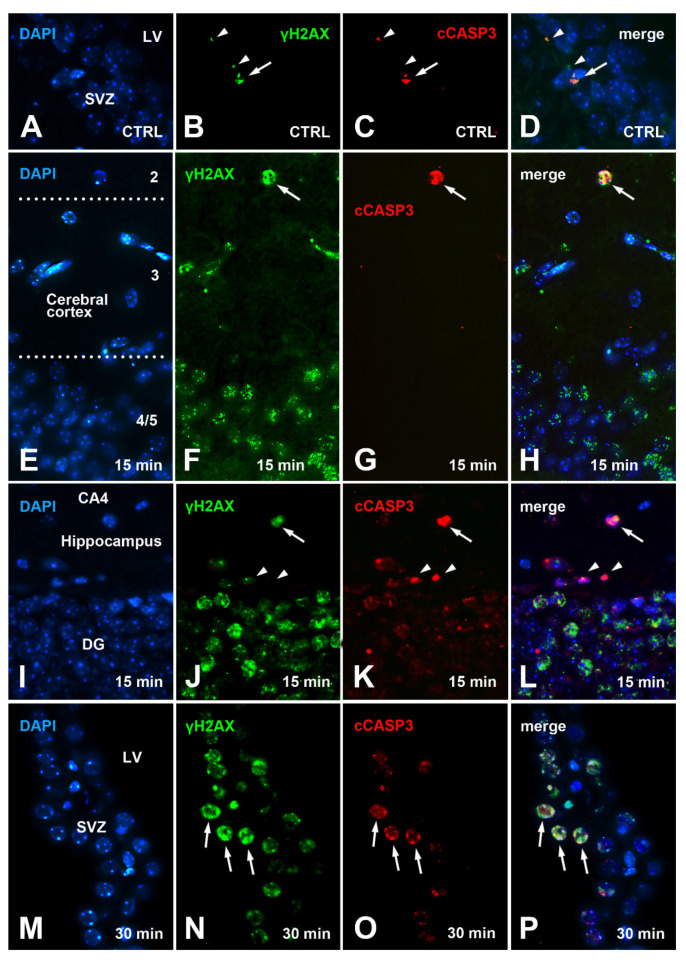
Exemplificative wide-field IMF images of γH2AX and cCASP3 immunoreactive cells in control and irradiated mice. (**A**–**D**) Localization of γH2AX and cCASP3 in a cell with apoptotic condensation (arrow) and apoptotic cell bodies (arrowheads) within the SVZ of a control not-irradiated mouse. (**E**–**H**) An apoptotic double-labeled cell (arrow) in layer 2 of the cerebral cortex of a mouse that survived 15 min following irradiation. (**I**–**L**) An apoptotic double-labeled cell (arrow) in the hippocampal CA4 field of a mouse surviving 15 min following irradiation. The arrowheads point to two condensed apoptotic bodies displaying cCASP3 immunoreactivity. Note that the one at left also shows a small focus of γH2AX positivity (panel J). (**M**–**P**) Several double-labeled cells in the SVZ after 30 min survival. The arrows indicate three γH2AX+cCASP3 positive cells. Abbreviations: CA = *Cornu Ammonis*; CTRL = control; DG = hippocampal dentate gyrus. Arabic numerals in € indicate cortical layers. Original magnifications 63×.

**Figure 8 biomedicines-09-01166-f008:**
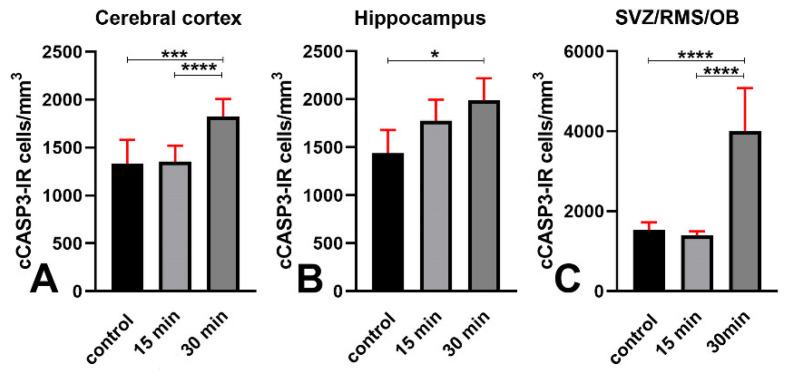
Volumetric density of cCASP3 immunoreactive cells in the cerebral cortex (**A**), hippocampus (**B**), and SVZ/RMS/OB (**C**) of control and irradiated mice. Data are grouped from two different sets of cCASP3+γH2AX double-labeling experiments with the Abcam and Upstate anti-γH2AX antibodies after check that cCASP3 immunostaining was not statistically different between the two experiments. Data are expressed as mean ± 95% CI. * 0.05 > *p* ≥ 0.01, *** 0.001 > *p* ≥ 0.0001, **** *p* < 0.0001. Bars are 95% CI. See Appendix G for numerical data of statistics.

**Figure 9 biomedicines-09-01166-f009:**
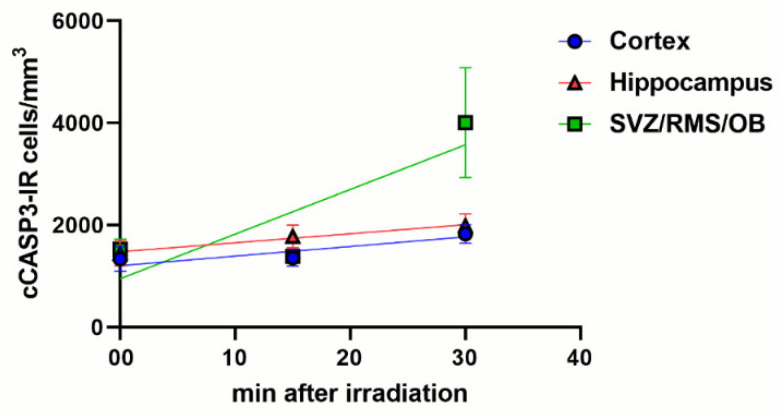
Linear regression curves of the volumetric density of cCASP3 immunoreactive cells in the cerebral cortex, hippocampus, and SVZ/RMS/OB in the function of the time of survival after X-ray irradiation. Note the different rates of response to X-rays in the SVZ/RMS/OB. Bars are 95% CI.

**Figure 10 biomedicines-09-01166-f010:**
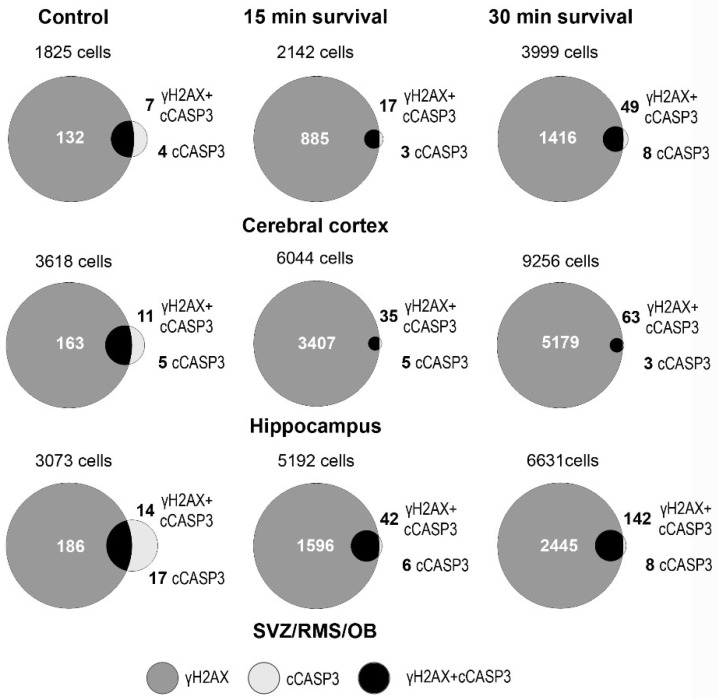
Visual representation of γH2AX + cCASP3 colocalization with quantitative Venn diagrams. The number at the top of each diagram is the total number of cells labeled with DAPI. Note that in control animals the cCASP3 immunoreactive cells represent a higher fraction of those labeled with γH2AX and that there is a relatively large population of cCASP3 single-labeled cells in these animals. Irradiation reduces the percentage of cCASP3 immunoreactive cells versus those labeled with γH2AX and leads to a drastic reduction of the cells singularly labeled for cCASP3 that almost disappear in the hippocampus and SVR/RMS/OB (see also Table 1).

**Figure 11 biomedicines-09-01166-f011:**
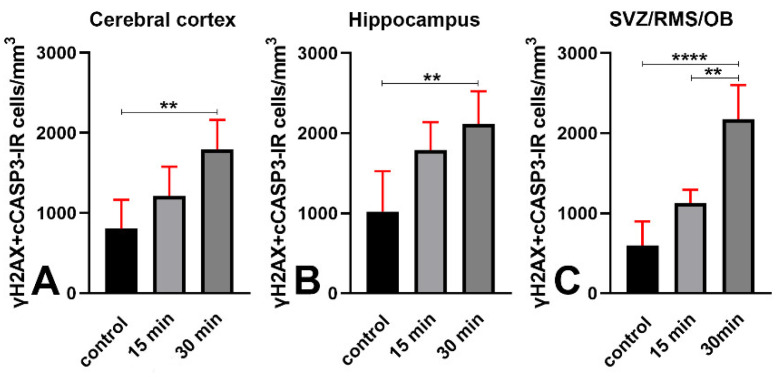
Effect of X-ray irradiation on the volumetric density of γH2AX+cCASP3 immunoreactive cells in the cerebral cortex (**A**), hippocampus (**B**) and SVZ/RMS/OB (**C**). Staining for γH2AX was carried out with the Abcam primary antibody. Data are expressed as mean ± 95% CI. ** 0.01 > *p* > 0.001, **** *p* < 0.0001. Bars are 95% CI. See Appendix H for numerical data of statistics.

**Figure 12 biomedicines-09-01166-f012:**
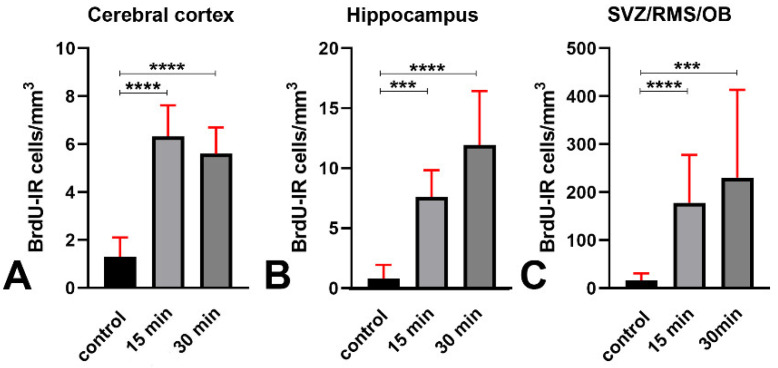
Effect of X-ray irradiation on the volumetric density of BrdU immunoreactive cells in the cerebral cortex (**A**), hippocampus (**B**) and SVZ/RMS/OB (**C**). Data are expressed as mean ± 95% CI *** 0.001 > *p* ≥ 0.0001, **** *p* < 0.0001. Bars are 95% CI. See Appendix I for numerical data of statistics.

**Figure 13 biomedicines-09-01166-f013:**
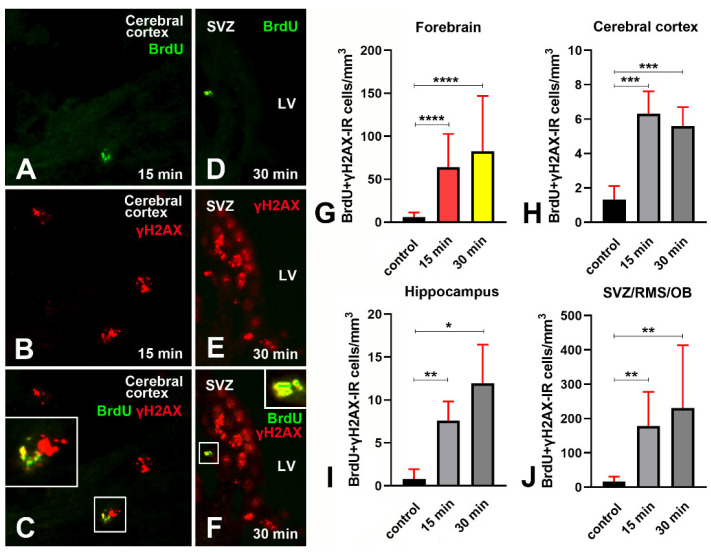
BrdU + γH2AX double-labeling experiments in the forebrain of control and irradiated mice. (**A**–**F**) exemplificative images of the pattern of immunostaining in the cerebral cortex after 15 min survival (**A**–**C**) and SVZ after 30 min survival (**D**–**F**). Note the pattern of immunostaining in foci for both molecules. Two double-labeled cells (rectangles) are shown at higher magnification in the inserts. Abbreviations: LV = lateral ventricle; SVZ = subventricular zone of the lateral ventricle. Original magnifications: 20×, inserts, 40×. (**G**–**J**) Quantitative analysis of the volumetric density of double-labeled cells, in the whole forebrain, cerebral cortex, hippocampus, and SVZ/RMS/OB. Data are expressed as mean ± 95% CI. * 0.05 > *p* ≥ 0.01, ** 0.01 > *p* ≥ 0.001, *** 0.001 > *p* ≥ 0.0001, **** *p* < 0.0001. Bars are 95% CI. See Appendix J for numerical data of statistics.

**Figure 14 biomedicines-09-01166-f014:**
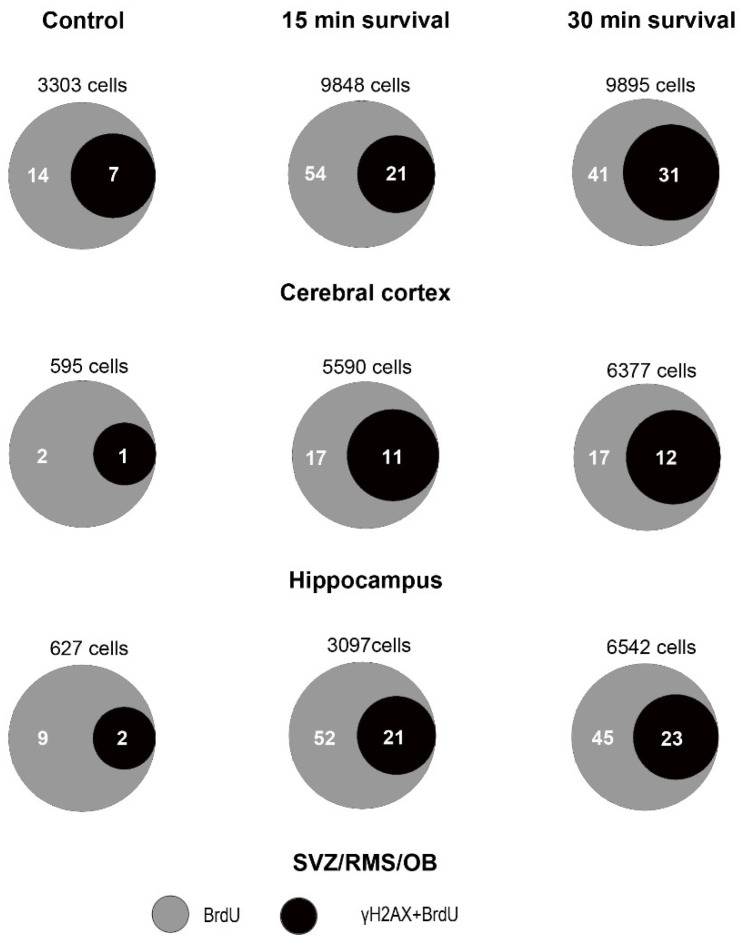
Visual representation with quantitative Venn diagrams of the data presented in Figure 13. Diagrams show the absolute numbers of BrdU single-labeled cells (gray) and BrdU + γH2AX double-labeled cells (black) after double-labeling experiments. Numbers outside the diagrams are the total number of DAPI-labeled nuclei. Numbers inside the diagrams are the absolute numbers of immunoreactive cells. The fraction of γH2AX single-labeled cells is not represented for diagram readability. Note that in all experimental settings, double-labeled cells represent a sizable fraction of BrdU+ cells and that this fraction increased with survival time after X-ray irradiation, at least in absolute terms.

**Figure 15 biomedicines-09-01166-f015:**
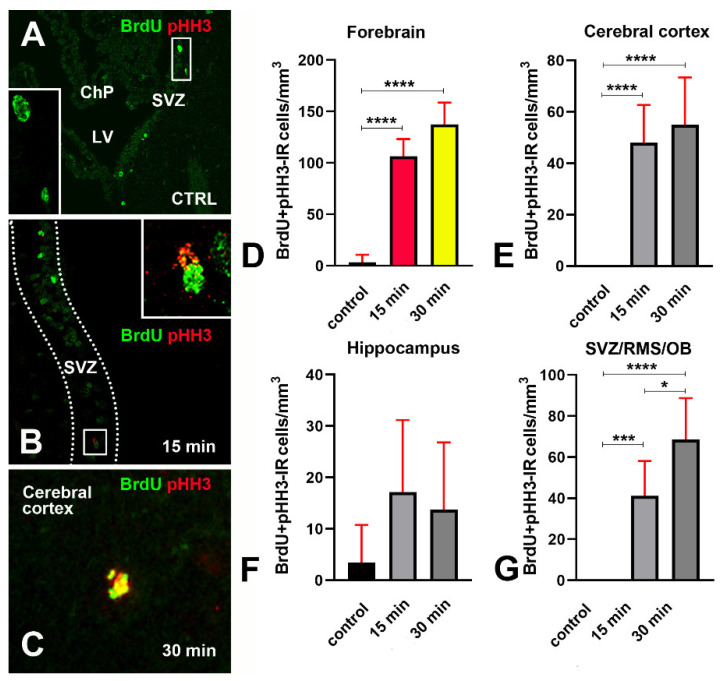
BrdU + pHH3 double-labeling experiments in the forebrain of control and irradiated mice. (**A**) exemplificative image of the pattern of immunostaining in the SVZ of a control nonirradiated mouse. Note that all immunoreactive cells are singularly labeled for BrdU (green) and the lack of double-labeled cells. The two BrdU+ cells in the rectangle are shown at higher magnification in the insert. (**B**) exemplificative image of the pattern of immunostaining in the SVZ of a mouse that survived 15 min after irradiation. Note that most immunoreactive cells are singularly labeled for BrdU (green). The two cells in the square appear at higher magnification in the insert. Note the presence of foci of colocalization (yellow) of the two labels, which are particularly evident in the top cell that is mainly reactive for pHH3 (red). (**C**) exemplificative image of the pattern of immunostaining in the cerebral cortex of a mouse that survived 30 min after irradiation. Note that the cell is double-labeled and displays clear foci of immunoreactivity for BrdU (green), pHH3 (red), or both (yellow). (**D**–**G**) Quantitative analysis of the volumetric density of double-labeled cells in the whole forebrain, cerebral cortex, hippocampus, and SVZ/RMS/OB. Abbreviations: ChP = choroid plexus; LV = lateral ventricle; SVZ = subventricular zone of the lateral ventricle. Original magnifications: (**A**,**B**) 20×; (**C**) and inserts, 40×. Data are expressed as mean ± 95% CI. * 0.05 > *p* ≥ 0.01, *** 0.001 > *p* ≥ 0.0001, **** *p* < 0.0001. Bars are 95% CI. See Appendix K for numerical data of statistics.

**Figure 16 biomedicines-09-01166-f016:**
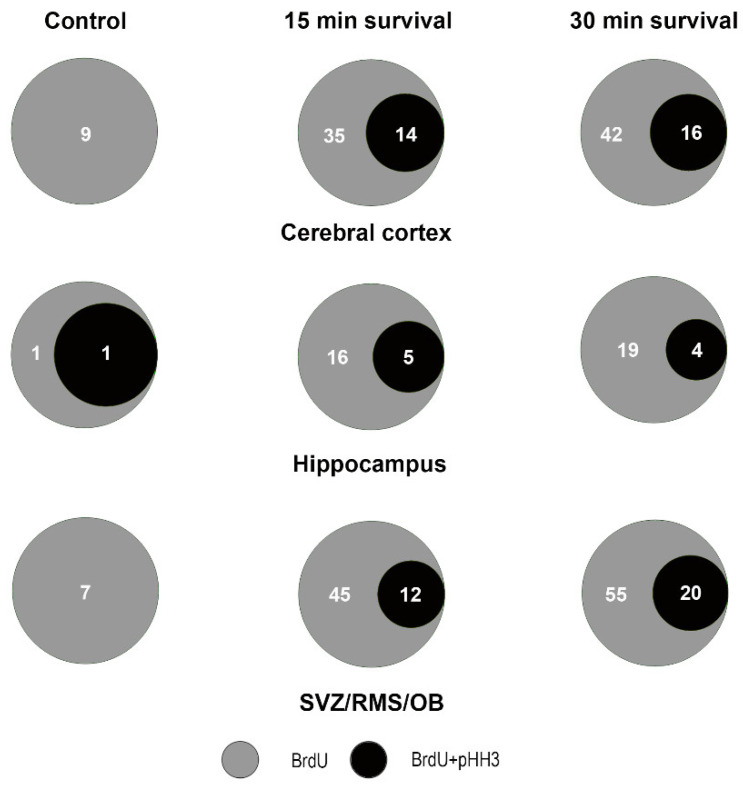
Visual representation with quantitative Venn diagrams of the data presented in Figure 15. Diagrams represent the total number of cells labeled with BrdU (gray), and BrdU+pHH3 (black) under different experimental conditions. Note that in all experimental settings, double-labeled cells represent a fraction of BrdU+ cells and that this fraction increases with survival time after X-ray irradiation, except in hippocampi.

**Table 1 biomedicines-09-01166-t001:** Predicted co-labeling percentage (PCP) and observed co-labeling percentage (OCP) of γH2AX+cCASP3 immunoreactive nuclei in the forebrain areas of control and irradiated mice. Note that data for cCASP3 are expressed as per mil numbers of cells. Therefore, calculation of PCP is made using the following formula: $\frac{\%\;\gamma H2\mathrm{AX}\times ‰\;\mathrm{cCASP}3}{10}$ based on the joint probability of event A (positivity for γH2AX) and event B (positivity for cCASP3) and considering the two events independent. In this case, computing is made by simply multiplying the probabilities of the two events: P(A∩B)=P(A)×P(B).

Forebrain Areas—Animal Groups	% γH2AX	‰ cCASP3	PCP (%)	OCP (%)
Cerebral cortex-Control	8.055	6.296	5.0714	68.75
Cerebral cortex-15 min survival	39.65	10.66	42.267	85.9
Cerebral cortex-30 min survival	35.97	14.55	52.336	89.17
Hippocampus-Control	4.106	4.394	1.804	68.75
Hippocampus-15 min survival	55.78	6.517	36.352	89.06
Hippocampus-30 min survival	56.22	7.079	39.798	95.31
SVZ/RMS/OB—Control	7.036	10.48	7.374	43.21
SVZ/RMS/OB-15 min survival	30.72	10.29	31.611	87.81
SVZ/RMS/OB-30 min survival	37.39	23,6	88.240	93.76

## Data Availability

The data that support the findings of this study are available from the corresponding author upon reasonable request.

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
