# Peer review of "Association of Caspase 3 Activation and H2AX γ Phosphorylation in the Aging Brain: Studies on Untreated and Irradiated Mice"

_biomedicines, 2021, doi:10.3390/biomedicines9091166_

Round 1

Reviewer 1 Report

The manuscript is clear and well-written. The experimental design is straightforward, enabling the authors to dissect the early molecular events of radiation exposure in the mouse brain. This study reconfirms the sequence of events leading to apoptosis in the dividing cells of the brain (neural stem cells) and other affected differentiated cell types. In my opinion, the manuscript is acceptable. I have only minor comments. 

1.There are formatting issues throughout the manuscript, including many one-sentence paragraphs.

2.The authors previously observed gamma-H2AX foci in the aging mouse brain. Have the authors considered retrotransposon activity/insertion during aging as the cause for DNA double-strand breaks? 

Gasior SL, Wakeman TP, Xu B, Deininger PL. The human LINE-1 retrotransposon creates DNA double-strand breaks. J Mol Biol. 2006 Apr 14;357(5):1383-93.

Erwin, J., Marchetto, M. & Gage, F. Mobile DNA elements in the generation of diversity and complexity in the brain. Nat Rev Neurosci 15, 497–506 (2014). 

3.The authors may want to elaborate on the significance of different cell types, brain regions, and the central nervous system's radiation toxicity response. How does it translate to humans undergoing radiation treatment for brain tumors? Is there a mechanism and strategy for injury reduction based on this study?

4.How do age and radiation affect the levels of gamma-H2AX and DDR in cells which make up the brain's blood vessels?

Author Response

Reviewer 1

Comments and Suggestions for Authors

The manuscript is clear and well-written. The experimental design is straightforward, enabling the authors to dissect the early molecular events of radiation exposure in the mouse brain. This study reconfirms the sequence of events leading to apoptosis in the dividing cells of the brain (neural stem cells) and other affected differentiated cell types. In my opinion, the manuscript is acceptable. I have only minor comments. 

Thank you very much for the appreciation of our work.

  1. There are formatting issues throughout the manuscript, including many one-sentence paragraphs.

We have formatted the manuscript and edited it for English as also suggested by Reviewer 2.

2.The authors previously observed gamma-H2AX foci in the aging mouse brain. Have the authors considered retrotransposon activity/insertion during aging as the cause for DNA double-strand breaks?

Gasior SL, Wakeman TP, Xu B, Deininger PL. The human LINE-1 retrotransposon creates DNA double-strand breaks. J Mol Biol. 2006 Apr 14;357(5):1383-93.

Erwin, J., Marchetto, M. & Gage, F. Mobile DNA elements in the generation of diversity and complexity in the brain. Nat Rev Neurosci 15, 497–506 (2014). 

Being not experts in the field of retrotrasposomes we had not considered this possibility. As the suggestion is very appealing, we have discussed it including the two above-indicated references, as follows (lines 841-848 of revised MS):

We do not know which exactly may be the source and type of DNA damage in un-treated mice. However, it would be interesting to speculate on the intervention of re-trotransposomes in the generation of the DSBs that may trigger the phosphorylation of H2AX in normal aging. Retrotransposomes are mobile DNA elements that can change their position within the genome, producing genomic diversity between neurons and in-tervening in certain neurological disorders [58]. Remarkably, the human LINE-1 re-trotransposon generates high levels of DNA DSBs in HeLa cells, as detected with γH2AX immunocytochemistry and single-cell gel electrophoresis [59].

3.The authors may want to elaborate on the significance of different cell types, brain regions, and the central nervous system's radiation toxicity response. How does it translate to humans undergoing radiation treatment for brain tumors? Is there a mechanism and strategy for injury reduction based on this study?

This is a very interesting comment and we thank the reviewer for it. We did not speculate on this issue in the original MS, as the study of the effect of irradiation was not the primary focus of our study. Yet, in the Discussion we have included a brief comment about the possible translation of our findings in radiation therapy as follows (lines 791-810 of revised MS):

Finally, we would like to comment briefly about the potential translational significance of our results. Patients undergoing whole brain radiation therapy suffer from a series of side effects, including a severe and irreversible cognitive decline [50,51]. Remarkably, the radiation-induced neurocognitive impairment is classically thought to be the con-sequence of multiple mechanisms (i.e., neuroinflammation, decreased neurogenesis, reduced proliferation of vascular and glial cells), chiefly (but not only) affecting the hippocampus [52].

Using a mouse model and γH2AX immunocytochemistry, very recently it was suggested that early life irradiation induced persistent DNA damage foci at later stages of life, resulting in more rapid structural and cognitive aging and shortened life expectancy [53]. Our results indicate that an additional mechanism of cognitive impairment in the aging brain may be represented by an early-onset increased susceptibility to apoptosis, which leads to widespread cell death in the neocortex and hippocampus, two structures deeply involved in cognitive functions. Death could affect not only the neurons [38], but also the glia and cells of the small blood vessels, which can proliferate and are responsible e.g., of the vascular cognitive impairment associated with subcortical small vessel disease [54]. This view is also supported by the efficacy of antiapoptotic drugs, such as memantine, for the prevention of radiation-induced cognitive decline [55]. Thus, therapeutic interventions to reduce the cognitive impact of radiation therapy might be supported by additional strategies in the future.

4.How do age and radiation affect the levels of gamma-H2AX and DDR in cells which make up the brain's blood vessels?

Again, this is a very interesting comment. However, we must say that we specifically excluded blood vessels (e.g., those in the choroid plexuses) from our analysis, as endothelial cells easily pick up BrdU as they can proliferate. Yet we have very briefly commented on this issue in the response to comment 3 (see text above).

Reviewer 2 Report

The article entitled “Association of Caspase 3 Activation and H2AX γ Phosphorylation in the Aging Brain: Studies on Untreated and Irradiated Mice” is a continuation of a previous study PMID: 24451138. Authors tried to find and describe the connection between caspase 3 and H2AX in vivo in irradiated and normal aging brain cells and proposed the following results: studied brain areas have a different vulnerability to irradiation, H2AX phosphorylation increases over time in the forebrain and is correlated with 53BP1 expression, irradiation causes DDR accompanied by DNA synthesis and mitotic response. The study is well-prepared; however, the manuscript could be improved regarding introduction and results’ description and their following explanation. Also, I recommend shortening the manuscript and using an English-editing service, as the style must be improved (together with numerous spelling mistakes) to be easy to follow and comprehend by future readers.

Below, minor remarks are listed point by point.

Abstract:

line 10: Please remove the first sentence.

line 16: Lack of 53BP1 explanation.

line 14-18: Please re-write and divide into two sentences.

Introduction

line 39-41: Please add abbreviations of ROS and BER.

line 39-42: Please divide into two sentences and describe potential biological consequences in more detail.

line 39-49: Please give a level (with appropriate references) of SSBs and DSBs present in the cell in order to underline the severity of the problem when such breaks occur.

line 54: Please give examples of in vitro studies other than neural cell lines – which cells were employed previously?

line 57: Please change to “of DNA damage” instead of “insults to the DNA integrity”

line 62: Please describe these few findings that you are referring to. If the research is scarce on the possible connections between H2AX and CASP3 it should be elucidated in more detail.

line 63: Please remove the second part of the sentence starting from “as detected…”

line 68-74: It is mentioned that previous studies examined mice ages 14.5 days up to 24 months. However, the comment on the results of 24-month-old mice is missing. I suggest to re-write this part and add more detail or shorten information about the study and focus on describing relevant results of this previous study.

line 89: Adult and old – mice? humans? Please clarify.

line 90: more important than what? Please clarify or re-write the sentence.

Methods:

line 135: Please give the numbers of mice in each group.

line 152: Change “see below” to specific reference e.g., see 2.3.2.

line 252: I believe that 2.7. is not necessary as a separate point, please add this information to the previous section.

Figure 1: The schematic is very helpful and informative, but I do believe it may be improved. The bars presenting DDR increase should be moved a little to increase space between the bar and mice icons above them – now, the first impression is that bars correspond to group 1 and 2, not 2 and 3 as intended. Also, an explanation of bars coloring should be placed within the figure – it would increase readability and shorten the figure caption, as a detailed description of the meaning of colors would not be needed. I suggest considering adding more details and moving this figure to the Introduction or discussion, as the figure is nice and very helpful to follow the experiment’s design and results of the study. 

Results:

The Results section, in my opinion, should be more concise and describe current data and results only.

I suggest that authors consider placing some numerical statistical data in a Table, as it might be easier to read and follow presented results.

Also, please place figures after paragraphs where they are first mentioned to increase readability.

Figure 3-8: I believe that the captions should be shortened, and the description of results and numerical data should be placed in the appropriate sections of “Results”.

line 458-472: Please make sure that comas are placed correctly instead of the point. Now, means are given as, for example, 316 thousand +- 35 thousand. Probably authors should correct these comas to points.

Figure 9: It seems that this figure is not mentioned in the text. If so, please add the reference, or remove it and correct the numbering of other figures.

Figure 11: Please use the same range on the vertical axis.

Table 1: Please address the difference between PCP and OCP in control groups. Why the difference is so substantial? Based on what did the authors made predictions that were so different than observed values?

line 601: Please specify the species you are referring to with “adult neurogenesis”.

Please comment on how the survival times were chosen and why there were no longer times tested.

Discussion:

Please make sure to refer in the whole discussion section to the figures/sections from the results when they are presented (e.g., line 806-821).

Also, I do believe that detailed descriptions of previously obtained results and conclusions should be removed and authors should focus on presenting and discussing currently obtained results as in the present form it is sometimes not clear whether past or current results are taken into consideration. In my opinion, it is sufficient to place a reference and only shortly mention previous experiments without describing them, only using them as a part of the discussion.

line 777-778: Is it rightly correlated that the extent of DNA damage may be assessed only by the number of DSBs? There are other types of damage to DNA non-DSBs. Please elucidate or re-write.

line 784: Please refrain from using the phrase “injured DNA”; change to damaged.

line 798 and 848: Please indicate the model organism of the mentioned study.

line 824: Please indicate used x-ray dosage.

line 902: Please re-write the sentence. It is not clear what are the authors referring to.

line 935-936: Please explain the abbreviations used.

line 974: Please provide references.

line 981: Authors note that studies on P60 mice should be interpreted with caution, but in the introduction and their previous study 2-month-old mice were discussed. Could authors comment on that?

line 995: Please refrain from phrases such as “very elegant study”.

line 985-996: Please remove or shorten to the maximum this fragment and only describe current results. If necessary, previous results may be mentioned, but should not be described.

line 997-1003: Please consider presenting these findings as a list for clarity.

line 1019: Please remove the parenthesis from (mouse).

Conclusions:

line 1044: Please elucidate on the type of damage that authors had in mind here.

Appendices:

Please re-check the numbering of the figures in Appendices, as they are not correct.

Please refrain from detailed descriptions of other groups' studies (e.g., remove lines 1180-1185). Authors should concentrate on describing obtained values and comparing them to literature data, not the other way around (describing literature data and shortly comparing them to presented results).

References:

Please revise and try to replace some references with newer ones as there are some great papers available in the field e.g., from the last 5 years.

English:

Please improve English, shorten subordinate sentences to increase readability, use past simple throughout the manuscript when referring to other studies or experiments (e.g., line 58 and 749), correct minor spelling and punctuation mistakes (such as x-ray to X-ray), abbreviations are given only in some figures’ captions, etc. I highly recommend using the professional English-editing service.

Below, a few minor errors are listed (please carefully revise the whole text):

line 54: “other” instead of “others”

line 56: Here is the first mention of caspase 3, the abbreviation is missing (it is given only later in line 61)

line 121: Please correct to “24-month-old”

line 208: Add “a” to “single labeled”.

line 257: Gamma sign is wrongly placed.

line 268: I believe it should be “control mice”. Please check throughout the manuscript whether the correct is “mouse” or “mice”, as it suggests that some experiments were performed on a single subject.

line 281: “4 in to 8%”? Please correct.

line 489: Please change “at 30 min” to “up to 30 min”.

line 609: Did the authors mean “few” or “a few”?

line 744: Please change “to disturbs” to “to the disturbances in DNA repair…”

line 781 and 843: “in accord”? Did the authors mean “in accordance”?

line 817: “awkward”? It means something causing difficulties. Maybe authors meant, for example, contrary or opposite? Please improve wording throughout the whole manuscript, as it may be very confusing for the reader. Also, as listed above, shortening sentences would improve the readers’ comprehension.

line 921: “were for the most neurons” – were present? What authors meant here?

line 960: “with and”? Please revise.

line 1021: “elderly” refers to humans, please revise.

line 1030: Please remove “an”

Author Response

Reviewer 2

The article entitled “Association of Caspase 3 Activation and H2AX γ Phosphorylation in the Aging Brain: Studies on Untreated and Irradiated Mice” is a continuation of a previous study PMID: 24451138. Authors tried to find and describe the connection between caspase 3 and H2AX in vivo in irradiated and normal aging brain cells and proposed the following results: studied brain areas have a different vulnerability to irradiation, H2AX phosphorylation increases over time in the forebrain and is correlated with 53BP1 expression, irradiation causes DDR accompanied by DNA synthesis and mitotic response. The study is well-prepared; however, the manuscript could be improved regarding introduction and results’ description and their following explanation. Also, I recommend shortening the manuscript and using an English-editing service, as the style must be improved (together with numerous spelling mistakes) to be easy to follow and comprehend by future readers.

We thank the reviewer for her/his appreciation of our work and very useful comments to improve its quality. We are also greatly indebted for the amount of time devoted to this very detailed revision.

Regarding MS shortage, we have done as much as we could to cope with this request. Yet reviewer 1 asked to add some comments on two aspects (role of transposomes and transaltional iimpotance of finding on X-rays) that we did not consider in the original MS. In addition some of the points listed below required to add a number of sentences with their relative references.

Below, minor remarks are listed point by point.

Abstract:

line 10: Please remove the first sentence.

Done

line 16: Lack of 53BP1 explanation.

Added

line 14-18: Please re-write and divide into two sentences.

Done

Introduction

line 39-41: Please add abbreviations of ROS and BER.

Done

line 39-42: Please divide into two sentences and describe potential biological consequences in more detail.

Done

line 39-49: Please give a level (with appropriate references) of SSBs and DSBs present in the cell in order to underline the severity of the problem when such breaks occur.

In response, we have added the following sentence (lines 46-52 of the revised MS)

SSBs and DSBs have different degrees of severity. DSBs are relatively rare but have a strong impact on neurodevelopment as they undermine the integrity of proliferating and differentiating cells, leading to an array of disorders from embryonic lethality to several forms of ataxia [3]. In contrast, SSBs, which are three orders of magnitude more frequent, are less severe as they undergo repair very quickly and, thus, are unlikely to cause developmental defects or microcephaly. Nevertheless, in post-mitotic neurons SSBs can be responsible of several forms of progressive neurodegeneration and cerebellar ataxia [3].

Another difference between SSBs and DSBs is their propensity to provoke apoptosis. line 54: Please give examples of in vitro studies other than neural cell lines – which cells were employed previously?

We have specified the main types of cells used in these studies at lines 62-63 of the revised MS.

line 57: Please change to “of DNA damage” instead of “insults to the DNA integrity”

Done

line 62: Please describe these few findings that you are referring to. If the research is scarce on the possible connections between H2AX and CASP3 it should be elucidated in more detail.

We have added the following paragraph (at lines 70-79 of the revised MS):

As regarding the connections between γH2AX and CASP3, the main effector protease in apoptosis [7], observations are very few in neurons and still controversial in other cell types. An initial study in non-neural HL60 cells has shown that induction of γH2AX in response to DNA lesions preceded apoptosis [8], but another survey has reported that H2AX was phosphorylated after activation of the apoptotic machinery in several human cell lines of different origins [9]. Studies in neurons, on the other hand, are not primarily focused onto the links between H2AX and apoptosis. They rather aim to assess the response of isolated cells or the intact brain to several DNA-damaging agents of heterogeneous nature such as e.g., hydrogen peroxide [10] and mifepristone [11] that have been reported to induce γH2AX with [11] or without [10] the concomitant activation of CASP3.

line 63: Please remove the second part of the sentence starting from “as detected…”

Done

line 68-74: It is mentioned that previous studies examined mice ages 14.5 days up to 24 months. However, the comment on the results of 24-month-old mice is missing. I suggest to rewrite this part and add more detail or shorten information about the study and focus on describing relevant results of this previous study.

We have shortened this part a bit, and better specified which data refer to 24 months old mice (lines 89-97 or revised MS).

line 89: Adult and old – mice? humans? Please clarify.

We have rephrased this into …in adult and old mammals, including humans (lines 105-106 of revied MS)

line 90: more important than what? Please clarify or re-write the sentence.

This was a misspell – changed in most (line 107 in the revised MS)

Methods:

line 135: Please give the numbers of mice in each group.

Done

line 152: Change “see below” to specific reference e.g., see 2.3.2.

Done

line 252: I believe that 2.7. is not necessary as a separate point, please add this information to the previous section.

Done

Figure 1: The schematic is very helpful and informative, but I do believe it may be improved. The bars presenting DDR increase should be moved a little to increase space between the bar and mice icons above them – now, the first impression is that bars correspond to group 1 and 2, not 2 and 3 as intended. Also, an explanation of bars coloring should be placed within the figure – it would increase readability and shorten the figure caption, as a detailed description of the meaning of colors would not be needed. I suggest considering adding more details and moving this figure to the Introduction or discussion, as the figure is nice and very helpful to follow the experiment’s design and results of the study. 

We thank very much the reviewer for the very useful comments that greatly improved the figure readability. The figure has been modified accordingly and details on the x-ray dosage and BrdU dosage/route of administration were added. According to suggestion, we also have moved the figure to Introduction.

Results:

The Results section, in my opinion, should be more concise and describe current data and results only.

We have tried to abbreviate this section as much as possible.

I suggest that authors consider placing some numerical statistical data in a Table, as it might be easier to read and follow presented results.

We have placed data in tables as much as possible, and moved tables in appendices to shorten the Results main text.

Also, please place figures after paragraphs where they are first mentioned to increase readability.

Done

Figure 3-8: I believe that the captions should be shortened, and the description of results and numerical data should be placed in the appropriate sections of “Results”.

We have followed these suggestions in full and placed most numerical data in Appendices. The remaining ones were added to main text (e.g. lines 288-290 of revised MS).

line 458-472: Please make sure that comas are placed correctly instead of the point. Now, means are given as, for example, 316 thousand +- 35 thousand. Probably authors should correct these comas to points.

We have double-checked this and commas were correctly placed. As these figures refer to volumetric cellular densities, they can reach values of hundred-thousand cells/mm3.

Figure 9: It seems that this figure is not mentioned in the text. If so, please add the reference, or remove it and correct the numbering of other figures.

We checked for the mention of figure in text (now at line 459).

Figure 11: Please use the same range on the vertical axis.

Done. Figure was corrected.

Table 1: Please address the difference between PCP and OCP in control groups. Why the difference is so substantial? Based on what did the authors made predictions that were so different than observed values?

Thank you for this very useful comment. Predictions were simply made based on the theory at the base of the probability of two events. We have tried to explain the strong difference between PCP and OTP adding some information to the Table legend, adding the indication of % to Table and to text (see lines 541-559 of revised MS).

line 601: Please specify the species you are referring to with “adult neurogenesis”.

Done

Please comment on how the survival times were chosen and why there were no longer times tested.

We have commented on this at beginning of section 4.1 (lines 677-679 of revised MS):

Irradiated mice were left to survive for 15- or 30-min as phosphorylation of H2AX reaches a peak within this interval to decline thereafter [19,20].

Discussion:

Please make sure to refer in the whole discussion section to the figures/sections from the results when they are presented (e.g., line 806-821).

We respectfully disagree with this suggestion as line numbers are removed in the published final version of the paper.

Also, I do believe that detailed descriptions of previously obtained results and conclusions should be removed and authors should focus on presenting and discussing currently obtained results as in the present form it is sometimes not clear whether past or current results are taken into consideration. In my opinion, it is sufficient to place a reference and only shortly mention previous experiments without describing them, only using them as a part of the discussion.

Thank you for this very useful comment. We have coped with it throughout discussion.

line 777-778: Is it rightly correlated that the extent of DNA damage may be assessed only by the number of DSBs? There are other types of damage to DNA non-DSBs. Please elucidate or re-write.

This sentence indeed refers to the response of cells to irradiation. Thus, in this, case we are correlating the effects of x-rays onto the DNA (DSBs) with the expression of γH2AX/foci. We have elucidated this as follows (lines 694-696 of revised MS):

Such an interpretation is coherent with the notion that the number of foci/nucleus after irradiation, as well as the amount of γH2AX is directly dependent on the number of DSBs and, thus, the extent of DNA damage [19].

line 784: Please refrain from using the phrase “injured DNA”; change to damaged.

Done

line 798 and 848: Please indicate the model organism of the mentioned study.

Done

line 824: Please indicate used x-ray dosage.

Done

line 902: Please re-write the sentence. It is not clear what are the authors referring to.

We have rephrased the sentence as follows (lines 851-855 of revised MS):

Regardless of the cause of DNA DSBs, the simultaneous cellular expression of γH2AX and 53BP1 in aged animals is coherent with the possibility that, at low-to-mid “natural” doses of DNA damage, the H2AX-mediated concentration of 53BP1 at DSBs is crucial to trigger a DDR. The latter would then directly result in apoptosis of postmitotic neurons or prevent the entry of damaged proliferating cells into mitosis [21,24].

line 935-936: Please explain the abbreviations used.

Done

line 974: Please provide references.

Done

line 981: Authors note that studies on P60 mice should be interpreted with caution, but in the introduction and their previous study 2-month-old mice were discussed. Could authors comment on that?

Thank you for this request. What we wanted to say is that studies simply using BrdU to demonstrate the occurrence of adult neurogenesis (i.e., detecting proliferation using BrdU only) in mice that at P60 are actually in between youth and adulthood are questionable as there may be remnants of proliferating cells still at this age which will disappear soon thereafter. We have tried to explain better our thoughts by adding the following sentence at the end of the paragraph (lines 928-929):

Indeed, it seems possible that BrdU positive cells at this age are simply the remnants of rare proliferating cells that will disappear soon thereafter.

line 995: Please refrain from phrases such as “very elegant study”.

Done

line 985-996: Please remove or shorten to the maximum this fragment and only describe current results. If necessary, previous results may be mentioned, but should not be described.

We have shortened to the maximum as follows (lines 931-936):

The interpretation of our present findings on the phosphorylation of H2AX in hippocampus and SVZ/RMS/OB is made more complex by the existence of adult neurogenesis in these two brain niches [60]. Our prior semi-quantitative study [13] indicated that the two neurogenic niches display a different behavior in relation to H2AX phosphorylation and left open the question whether activation of H2AX was related to the occurrence of DNA DSBs during cell division, apoptosis or, simply, to physiological activity, as demonstrated by others in hippocampus [69].

line 997-1003: Please consider presenting these findings as a list for clarity.

Done

line 1019: Please remove the parenthesis from (mouse).

Done

Conclusions:

line 1044: Please elucidate on the type of damage that authors had in mind here.

We have added the following at end of point 3 (line 988):

…naturally occurring endogenous DNA damage, directly in the form of DSBs or unrepaired SSBs that are converted to DSBs.

Appendices:

Please re-check the numbering of the figures in Appendices, as they are not correct.

Sorry about that - Done

Please refrain from detailed descriptions of other groups' studies (e.g., remove lines 1180-1185). Authors should concentrate on describing obtained values and comparing them to literature data, not the other way around (describing literature data and shortly comparing them to presented results).

Done

References:

Please revise and try to replace some references with newer ones as there are some great papers available in the field e.g., from the last 5 years.

We respectfully disagree, at least partly, with this remark since we believe it is important to quote the original papers rather than more recent reports that may often be only confirmative of the original findings. This also because the link of γH2AX and DDR was elucidated only in 1998.

Yet, we made a PubMed search with the following string “γH2AX AND neurons” and it resulted in 79 articles without filtering. If a filter is applied for the last 5 years articles are 35. Of these the most interesting is surely Weyemi et al. Histone H2AX promotes neuronal health by controlling mitochondrial homeostasis. Proc Natl Acad Sci USA. 2019, 116:7471-7476. Two other papers could be somewhat of interest i.e., Tang et al. Spatiotemporal dynamics of γH2AX in the mouse brain after acute irradiation at different postnatal days with special reference to the dentate gyrus of the hippocampus. Aging (Albany NY). 2021 13:15815-15832, which however analyses the effect of irradiation in postnatal mice (up to three weeks); and Loi et al. Increased DNA Damage and Apoptosis in CDKL5-Deficient Neurons. Mol Neurobiol. 2020 57:2244-2262, which addresses the intervention of γH2AX in CDKL5 deficiency disorder (CDD) using human neuroblastoma SH-SY5Y and CRISPR/Cas9-mediated genome editing, an issue very far from the focus of our work.

Then if “aging” was added to the search string we ended up with 7 papers most of which already present in the previous search and none of primary relevance to this study.

Having said so we have referred to the papers by Weyemi at al. in Introduction and Tang et al. in Discussion.

If this reviewer has some other specific suggestions, we will be happy to include other references as well.

English:

Please improve English, shorten subordinate sentences to increase readability, use past simple throughout the manuscript when referring to other studies or experiments (e.g., line 58 and 749), correct minor spelling and punctuation mistakes (such as x-ray to X-ray), abbreviations are given only in some figures’ captions, etc. I highly recommend using the professional English-editing service.

Done

Below, a few minor errors are listed (please carefully revise the whole text):

Thank you very much for spotting these out.

line 54: “other” instead of “others”

Corrected

line 56: Here is the first mention of caspase 3, the abbreviation is missing (it is given only later in line 61)

Moved abbreviation here

line 121: Please correct to “24-month-old”

Corrected

line 208: Add “a” to “single labeled”.

We do not understand this, as the original sentences was …carried out in single labeled sections

line 257: Gamma sign is wrongly placed.

Corrected

line 268: I believe it should be “control mice”. Please check throughout the manuscript whether the correct is “mouse” or “mice”, as it suggests that some experiments were performed on a single subject.

Thanks for this comment. Specifically, sentence at this line is correct as such since we are referring to the picture taken from one mouse. We have checked and amended this throughout the manuscript when necessary.

line 281: “4 in to 8%”? Please correct.

Corrected as 4 to 8%

line 489: Please change “at 30 min” to “up to 30 min”.

Corrected

line 609: Did the authors mean “few” or “a few”?

Corrected in a few

line 744: Please change “to disturbs” to “to the disturbances in DNA repair…”

Corrected

line 781 and 843: “in accord”? Did the authors mean “in accordance”?

Yes, we have amended this

line 817: “awkward”? It means something causing difficulties. Maybe authors meant, for example, contrary or opposite? Please improve wording throughout the whole manuscript, as it may be very confusing for the reader. Also, as listed above, shortening sentences would improve the readers’ comprehension.

We have changed it to “odd” and we had the manuscript thoroughly edited for English.

line 921: “were for the most neurons” – were present? What authors meant here?

We have changed it in “most γH2AX immunoreactive cells were neurons”

line 960: “with and”? Please revise.

We have corrected the typo in “with an”

line 1021: “elderly” refers to humans, please revise.

Revised and changed in old

line 1030: Please remove “an”

Done